# TEXTUAL STEERING VECTORS CAN IMPROVE VISUAL UNDERSTANDING IN MULTIMODAL LARGE LANGUAGE MODELS

## ABSTRACT

Steering methods have emerged as effective tools for guiding large language models' behavior, yet multimodal large language models (MLLMs) lack comparable techniques due to recency and architectural diversity. Inspired by this gap, we demonstrate that steering vectors derived solely from text-only LLM backbones can effectively guide their multimodal counterparts, revealing a novel cross-modal transfer that enables reuse of existing interpretability tools. Using community-standard methods—Sparse Autoencoders (SAE), Mean Shift, and Linear Probing—we systematically validate this transfer effect across diverse MLLM architectures and visual reasoning tasks. Text-derived steering consistently enhances multimodal performance, with mean shift achieving up to +7.3% improvement in spatial relationship accuracy and +3.3% in counting accuracy on CV-Bench, and exhibits strong generalization to out-of-distribution datasets. These results highlight textual steering vectors as a powerful, efficient mechanism for enhancing grounding in MLLMs with minimal additional data collection and computational overhead.

## 1 INTRODUCTION

Steering large language models (LLMs) via their internal representations has emerged as a lightweight, interpretable paradigm for eliciting safe and controllable behavior (Li et al., 2023a; Turner et al., 2023; Sharkey et al., 2025, *inter alia.*). However, similar steering approaches have not yet gained prominence for *multimodal large language models* (MLLMs). This is in part due to their relative recency, as well as the heterogeneity of their architectures compared to text-only LLMs. Moreover, many steering methods assume access to a dataset of contrast pairs (Marks and Tegmark, 2023) to construct steering vectors, which may not be readily available for multimodal inputs.

Our key finding is that internal representations from a text-only LLM backbone retain their steering effectiveness even after multimodal adaptation. This transfer effect enables a new multimodal steering paradigm that is agnostic to architecture and does not require specialized multimodal data. Importantly, it also allows us to directly repurpose steering methods originally developed for text-only models—such as Sparse Autoencoders (SAEs), Mean Shift, and Linear Probing—without modality-specific modifications. This bridges the mature ecosystem of text-based steering (McGrath et al., 2024; Durmus et al., 2024; Hanna et al., 2025) with the emerging space of multimodal models, providing a lightweight and interpretable pathway for enhancing multimodal reasoning.

Building on this insight, we propose a plug-and-play framework for multimodal steering. We extract steering vectors from text-only LLM backbones using established techniques and then apply them to the hidden states of their multimodal counterparts. This approach leverages the existing toolbox of steering methods, which have been extensively studied and evaluated in the text domain, to ensure accessibility and broader applicability for multimodal research. In contrast, developing new multimodal-specific steering methods would require both specialized datasets and bespoke implementations, which can be difficult to adapt across different modalities and fusion architectures.

We evaluate our approach across multiple open-weight MLLMs and a broad suite of visual reasoning tasks. Our method consistently outperforms prompting baselines–for example, Mean Shift achieves up to +7.3% improvement in spatial relationship accuracy on CV-Bench. Notably, while direct prompting is effective for controlling *text-only* LLM behavior (Wu et al., 2025), it provides little

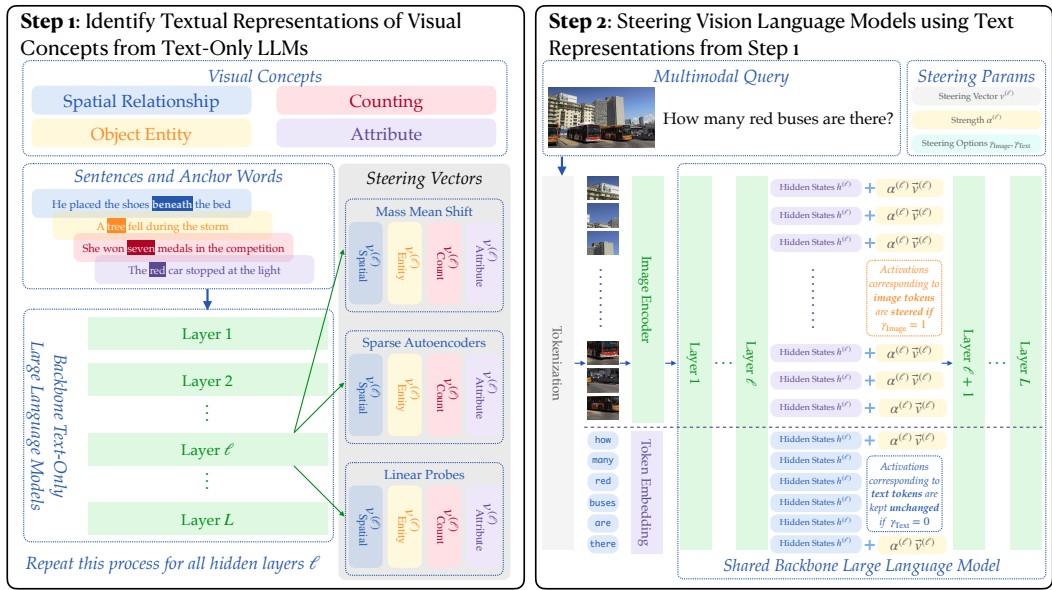

Figure 1: Overview of our steering methodology. Given an MLLM with a text-only LLM backbone and an image-bound prompt, we first identify the required visual concept (*e.g*, spatial relationships, counting). For each hidden layer $\ell$, we then extract corresponding steering vectors from the underlying LLM using Mean Shift, Linear Probing, or Sparse Autoencoders. Finally, we apply these vectors to image tokens, text tokens, or both, controlled by parameters $\gamma_{\text{Image}}$ and $\gamma_{\text{Text}}$.

benefit for multimodal reasoning. We also compare against LoRA fine-tuning: although LoRA achieves stronger in-distribution accuracy, it exhibits limited out-of-distribution generalization and lacks the lightweight and interpretability advantages of steering. Our contributions are as follows:

- We introduce a plug-and-play multimodal steering method built directly on existing LLM representation-based techniques.

- We identify a novel transfer effect: representations from the text-only LLM backbone remain effective for steering its multimodal counterpart, even after vision-language post-training.

- We demonstrate consistent performance gains across multiple MLLMs and task categories. Importantly, we also show that textual steering vectors could generalize to out-of-distribution test sets and demonstrate significant performance gains.

## 2 RELATED WORKS

**Representation-Based Steering** methods are an effective family of methods for steering LLMs, often in two stages. First, they identify model components that influence target behaviors, using probing directions (Li et al., 2023a; Zou et al., 2023), activation differences (Li et al., 2023a; Turner et al., 2023; Panickssery et al., 2023; Marks and Tegmark, 2023; Lee et al., 2024), or lifted monosemantic features via SAEs (Lieberum et al., 2024b; Gao et al., 2025; Templeton et al., 2024; Marks et al., 2025) and their variants (Dunefsky et al., 2024), among other techniques. Second, they adjust steering hyperparameters to balance desiderata such as truthfulness (Lin et al., 2022; Hernandez et al., 2023; Li et al., 2023a), helpfulness (Zou et al., 2023), and quality.

While widely studied in LLMs, applying activation intervention to MLLMs remains elusive. To our knowledge, the only such effort is the VTI method (Liu et al., 2025), which extends LLM steering pipelines by constructing intervention vectors from paired multimodal inputs and applying them to both visual and textual representations. In contrast, we show that interventions vectors constructed solely from text inputs in the unimodal LLM can influence the MLLM's multimodal behavior. This result highlights an underexplored form of cross-modal transfer enabled by the preserved semantics (Lieberum et al., 2024b) of the text backbone.

**Shared Semantics** refer to the representations unifying heterogeneous modalities of the same content, as identified across languages in multilingual LLMs (Artetxe et al., 2019; Wendler et al., 2024; Wu et al., 2024) and text/vision inputs in multimodal models (Huh et al., 2024; Luo et al., 2024; Wu et al., 2024). Our work studies the transfer of steering effect across different modalities and training stages. Concurrently, Papadimitriou et al. show that SAE features co-activate across multimodal inputs, while our work explores how such shared features can be exploited to steer MLLMs.

**Multimodal Large Language Models** are commonly developed by endowing a backbone LLM with visual processing components and fine-tuning on multimodal datasets, with some exceptions still pretrained from scratch (Team, 2024a; OLMo et al., 2024; Chen et al., 2025). Using an LLM backbone typically involves projecting the outputs of an image encoder (Dosovitskiy et al., 2020; Zhai et al., 2023) to the same dimension as the underlying LLM by an MLP, and concatenating the resulting image/text tokens as input to the LLM. The model can then be finetuned on multimodal data, possibly with frozen layers (*e.g.*, in the LLM) to preserve pretrained knowledge.

## 3 TOY EXAMPLE

To demonstrate that textual representations can effectively intervene in visual understanding, we conduct a simple color perception experiment using GemmaScope (Lieberum et al., 2024a) for `Gemma-2-9B` for feature extraction and `PaliGemma2-10B-mix-448` (Beyer et al., 2024) as our target model. We present the model with a yellow-orange image (whose RGB hex code is #FFB400) and manipulate its perception by intervening in the hidden representations. Specifically, we obtain the normalized red vector from GemmaScope and we add this vector to the hidden states of image tokens at layer 20 as follows: $h'_{\text{image}} = h_{\text{image}} + \alpha \cdot v_{\text{red}}$, where $\alpha$ is the scale factor controlling intervention strength. Figure 2 shows how increasing the scale factor shifts perception along a color spectrum: initially yellow-orange dominates, then orange peaks at scale factor 50, and finally red becomes dominant beyond scale factor 75. This demonstrates that textual

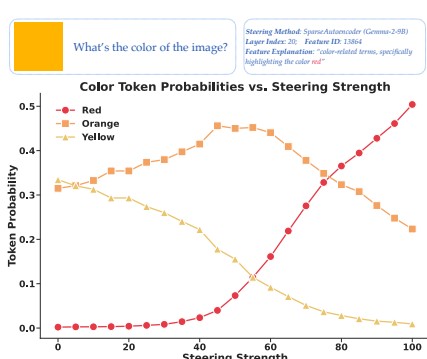

Figure 2: Effect of steering strength on color token probabilities.

features can integrate with and modify visual understanding, supporting our hypothesis of unified cross-modal representations within these models. We include more color examples in Appendix B.

## 4 METHODS

Building on our demonstration that textual representations can effectively steer visual understanding, we now explore systematic approaches to improve MLLMs' visual reasoning. Despite their growing success, MLLMs still struggle with seemingly simple visual queries—miscounting objects, confusing spatial relationships, and mishandling compositional prompts (Fu et al., 2024b). When the same problems are posed in pure text, foundation models perform far better (Fu et al., 2024a).

This observation motivates our central question: *Can existing steering mechanisms for textual representations rectify the shortcomings of MLLMs?* A promising remedy is steering vectors: compact directions in activation space that encode specific concepts. By adding these vectors to hidden representations at inference time as $h'_{\text{target}}{}^{(\ell)} = h_{\text{target}}^{(\ell)} + \alpha \cdot v^{(\ell)}$, we can amplify the model's internal representation of desired concepts without parameter updates. The optimal layer $\ell^*$ and scale $\alpha^*$ are found via grid search. We use three established methods—Sparse Autoencoders (SAE), Mean Shift, and Linear Probing—to extract vectors $v^{(\ell)}$ from text-only LLM backbones, demonstrating broad applicability of cross-modal transfer while ensuring accessibility and reproducibility.

### 4.1 DATASET CONSTRUCTION FOR STEERING VECTOR EXTRACTION

To extract high-level textual representations for visual concepts, we identify four important taxonomies for static images: spatial relationship, counting, attribute, and entity (Huang et al., 2023; Lin et al., 2024; Fu et al., 2025). For each visual concept, we curate small sets of sentence-anchor

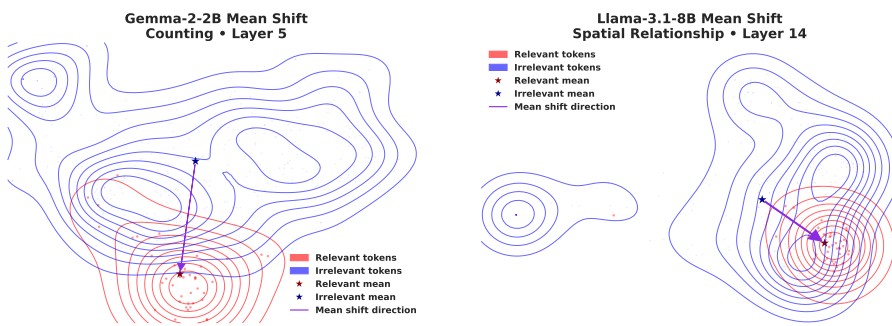

Figure 3: **Left:** Mean Shift method for counting features in `Gemma-2-2B`. The direction points from mean control token states to mean counting-related token states. **Right:** Spatial relationship features for `Llama-3.1-8B`. Activations projected to 2D for visualization.

pairs, where each pair contains a sentence exhibiting the visual concept and the specific anchor word representing that concept. These sentence-anchor pairs serve as the foundation for all three steering vector extraction methods. Examples are provided in Table 4 in Appendix A.1.

## 4.2 INTERPRETABLE STEERING VECTOR EXTRACTION METHODS

**Sparse Autoencoders (SAE).** Sparse Autoencoders reconstruct the activations of an LLM's hidden layer using an MLP with a single hidden layer and a sparsity penalty on the hidden layer. More precisely, let $x = h^{(\ell)}(t) \in \mathbb{R}^D$ be the model activations for a token $t$ at layer $\ell$ in an LLM. A SAE reconstructs $x$ as $\hat{x} = b_{\text{dec}} + \sum_{i=1}^{F} f_i(x) W_{\cdot,i}^{\text{dec}}$, where $b_{\text{dec}} \in \mathbb{R}^D$ and $W^{\text{dec}} \in \mathbb{R}^{D \times F}$ are learned decoder weights, and $f_i(x)$ is the activation corresponding to feature $i$. Feature activations are computed using learned encoder weights $W^{\text{enc}} \in \mathbb{R}^{F \times D}$ and $b^{\text{enc}} \in \mathbb{R}^F$ as $f_i(x) = \sigma\left(W_{i,\cdot}^{\text{enc}} x + b_i^{\text{enc}}\right)$, where $\sigma$ denotes an activation function of choice, *e.g.*, ReLU or JumpReLU.

The model is trained by minimizing the loss function $L = \mathbb{E}_x \left[ \|x - \hat{x}\|_2^2 + \lambda \sum_{i=1}^{F} f_i(x) \|W_{\cdot,i}^{\text{dec}}\|_2 \right]$, *i.e.*, $L_2$-reconstruction error and $L_1$-regularization on feature activations. In this formulation, unit-normalized decoder weight vectors $v_i^{(\ell)} := \frac{W_{\cdot,i}^{\text{dec}}}{\|W_{\cdot,i}^{\text{dec}}\|_2}$ serve as feature directions and $\alpha_i^{(\ell)}(t) := f_i(h^{(\ell)}(t)) \|W_{\cdot,i}^{\text{dec}}\|_2$ as the activation strength of $v_i^{(\ell)}$ on token $t$.

We leverage existing pretrained SAEs—GemmaScope (Lieberum et al., 2024b) for Gemma-2 models and LlamaScope (He et al., 2024) for Llama-3.1-8B. We emphasize that training SAEs is computationally expensive, and a key advantage of our approach is leveraging existing interpretability infrastructure without additional training costs. Using our sentence-anchor pairs, we identify features with high activations on anchor words. We then verify their relevance to the target visual concepts and average these relevant feature vectors to create a single steering vector for each visual concept at each layer. Additional details are provided in Appendix A.1.

**Mean Shift.** This method identifies feature directions by computing activation differences, as shown in Figure 3, showing surprising effectiveness for LLM steering (Marks and Tegmark, 2023; Wu et al., 2025). For each taxonomy $\mathcal{T}$ and layer $\ell$, using sentence-anchor pairs $\{(s_1, w_1), \ldots, (s_K, w_K)\}$, we compute the mean shift vector $m_{\mathcal{T}}^{(\ell)} = \frac{1}{K} \sum_{j=1}^{K} h^{(\ell)}(w_j) - \frac{1}{|\mathcal{S}_{\neg \mathcal{T}}|} \sum_{t \in \mathcal{S}_{\neg \mathcal{T}}} h^{(\ell)}(t)$, where $h^{(\ell)}(w_j)$ represents the residual stream activation of the anchor word $w_j$ at layer $\ell$ and $\mathcal{S}_{\neg \mathcal{T}}$ is a control set of non-anchor tokens from the same sentences. We refrain from normalizing the vector $m_{\mathcal{T}}^{(\ell)}$, preserving its magnitude relative to the original hidden states.

**Linear Probing.** We train a linear classifier distinguishing anchor word activations from control tokens on the $\ell$-th layer of a model (Alain and Bengio, 2016; Park et al., 2024). As the hidden state dimensionality often exceeds our sample size ($K < D$), we first project to dimension $d < K$ using PCA. With $Q \in \mathbb{R}^{d \times D}$ as the PCA matrix, the probe separates $\{h^{(\ell)}(w_j) Q^\top\}_{j \leq K}$ and $\{h^{(\ell)}(t) Q^\top\}_{t \in \mathcal{S}_{\neg \mathcal{T}}}$, where $\{(s_1, w_1), \ldots, (s_K, w_K)\}$ are the sentence-anchor pairs for concept $\mathcal{T}$

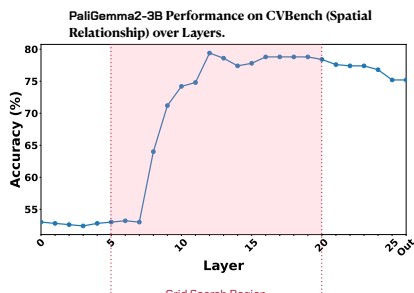

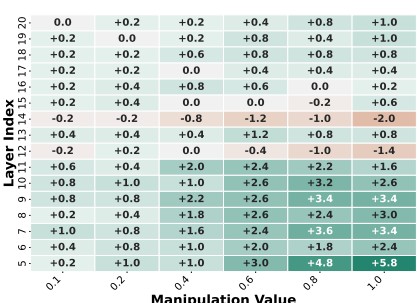

(a) We zero out models' attention to image tokens after layer $\ell$ and measure model performance. This reveals when visual information is processed and allows efficient grid search.

(b) Grid search on `PaliGemma2-3B` to locate the best $(\ell^*, \alpha^*)$ for steering the model's spatial reasoning abilities. In this case, $\ell^* = 5$ and $\alpha^* = 1.0$.

Figure 4: Efficient Grid Search with `PaliGemma2-3B` on the Spatial Relationship Task.

and $\mathcal{S}_{\neg\mathcal{T}}$ is our control set. The learned normal vector $v \in \mathbb{R}^d$ (pointing toward taxonomy-relevant points) yields the final steering vector $v' = Q^\top v$. We use $d = K/2$ in practice.

**Prompting Baseline.** Like our steering methods, prompting represents an interpretable approach that no parameter updates, and it has displayed impressive steering abilities in text-only domains (Wu et al., 2025). For a given taxonomy $\mathcal{T}$, we generate a prompt meant to enhance an MLLM's visual reasoning ability with respect to $\mathcal{T}$ as follows: We first curate a collection of 96 prompts of varying lengths by instructing `GPT-4o` to generate prompts that guide the model to reason with respect to $\mathcal{T}$, similar to the LLM-based prompt generation in AxBench (Wu et al., 2025), and then select the best-performing prompt via grid search on training data. Refer to Appendix A.2 for further detail.

## 5 STEERING IMPROVES MULTIMODAL LLMS

Having established in Section 3 that textual steering vectors applied to non-output tokens can alter the behavior of MLLMs, we now investigate whether the textual steering vectors we identified in Section 4.2 can *improve* visual understanding in MLLMs when applied to intermediate representations.

### 5.1 SETUP

**Models.** We investigate PaliGemma2 models with 3B and 10B parameters (`PaliGemma2-3B-mix-448` and `PaliGemma2-10B-mix-448`, referred to as `PaliGemma2-3B` and `PaliGemma2-10B`) and Idefics3-8B-Llama3. These models differ architecturally: PaliGemma2 adopts prefix-LM masking where image tokens and textual instructions are cross-attended, while Idefics3 is fully autoregressive following LLaVA. Steering vectors are extracted from their respective text-only backbones: `Gemma2-2B`, `Gemma2-9B`, and `Llama-3.1-8B`.

**Dataset.** We use CV-Bench (Tong et al., 2024) with 4 sub-categories: `Count`, `Relation`, `Distance`, and `Depth`, totaling 2,638 data points. Each sub-category contains around 700 samples, split into 500-600 training samples for grid search and 150 for testing.

**Grid Search.** We identify optimal injection layer $\ell$ and scale factor $\alpha$ via grid search on the training split. For each $(\ell, \alpha)$ pair, we intervene as $h'_{\text{target}}(\ell) = h_{\text{target}}(\ell) + \alpha v^{(\ell)}$ and select $(\ell^*, \alpha^*) = \arg\max_{\ell, \alpha} \text{Acc}(\ell, \alpha)$. We use $\mathcal{A} = \{0.1, 0.2, 0.4, 0.6, 0.8, 1.0\}$ for unnormalized vectors (MeanShift). For normalized vectors (SAE, Probe), we use $\{10, 20, 30, 40, 50, 60\}$ on PaliGemma2 models and $\{0.2, 0.4, 0.6, 0.8, 1.0, 1.2\}$ on Idefics3 due to smaller hidden state norms. We set $\mathcal{I}$ to be the middle layers, where we observe the learning from image tokens is predominantly happening (see Figure 4a): $\{5, 6, \ldots, 20\}$ for `PaliGemma2-3B` and `Idefics3-8B-Llama3`, and $\{15, 16, \ldots, 30\}$ for `PaliGemma2-10B`. Notably, we never steer output tokens, focusing on internal representations.

### 5.2 RESULTS

Table 1 presents a comparative analysis of three different models, `PaliGemma2-3B`, `PaliGemma2-10B`, and `Idefics3-8B-Llama3`, on tasks related to spatial relationships and counting in CV-Bench.

| MODEL | INTERVENTION TOKENS | | RELATION | | | COUNT | | |
|---|---|---|---|---|---|---|---|---|
| | TEXT | IMAGE | SAE | PROBE | MEANSHIFT | SAE | PROBE | MEANSHIFT |
| PaliGemma2-3B | — | | | 76.0 | | | 59.3 | |
| | ✓ | | 82.0 (+6.0)* | 77.3 (+1.3) | 83.3 (+7.3)* | 60.0 (+0.7) | 62.0 (+2.7) | 60.0 (+0.7) |
| | | ✓ | 78.7 (+2.7) | 76.7 (+0.7) | 78.7 (+2.7)* | 62.0 (+2.7)* | 60.7 (+1.3) | 62.0 (+2.7) |
| | ✓ | ✓ | 81.3 (+5.3)* | 78.7 (+2.7) | 81.3 (+5.3)* | 62.7 (+3.3)* | 62.0 (+2.7)* | 62.0 (+2.7)* |
| | Prompting | | | 76.7 (+0.7) | | | 60.0 (+0.7) | |
| PaliGemma2-10B | — | | | 79.3 | | | 63.3 | |
| | ✓ | | 78.7 (−0.7) | 77.3 (−2.0) | 83.3 (+4.0)* | 63.3 (+0.0) | 62.7 (−0.7) | 64.0 (+0.7) |
| | | ✓ | 79.3 (+0.0) | 79.3 (+0.0) | 78.7 (−0.7) | 63.3 (+0.0) | 63.3 (+0.0) | 64.7 (+1.3) |
| | ✓ | ✓ | 78.7 (−0.7) | 78.0 (−1.3) | 83.3 (+4.0)* | 64.0 (+0.7) | 63.3 (+0.0) | 63.3 (+0.0) |
| | Prompting | | | 76.7 (−2.7) | | | 63.3 (+0.0) | |
| Idefics3-8B-Llama3 | — | | | 73.3 | | | 59.3 | |
| | ✓ | | 76.0 (+2.7) | 78.0 (+4.7)* | 80.0 (+6.7)* | 58.7 (−0.7) | 58.0 (−1.3) | 60.0 (+0.7) |
| | | ✓ | 78.0 (+4.7)* | 72.7 (−0.7) | 76.7 (+3.3) | 60.0 (+0.7) | 59.3 (+0.0) | 60.7 (+1.3) |
| | ✓ | ✓ | 77.3 (+4.0)* | 78.7 (+5.3)* | 80.7 (+7.3)* | 62.0 (+2.7)* | 60.0 (+0.7) | 60.7 (+1.3) |
| | Prompting | | | 75.3 (+2.0) | | | 58.7 (−0.7) | |

Table 1: **Textual Steering Vectors Improve Multimodal LLMs' Visual Understanding**. Task-specific textual steering vectors reliably improve both spatial relation and counting performance across models. Stars ($\star$) denote statistically significant improvements ($p < 0.05$).

The performance is evaluated with and without intervention tokens (text, image, or both) and across different steering methods (SAE, Probe, MeanShift, and Prompting).

**Steering Interventions Prove Effective.** Table 1 demonstrates that steering interventions, especially MeanShift, consistently improve model performance on spatial relationship and counting tasks over baseline levels. For instance, PaliGemma2-3B's "Relation" accuracy with MeanShift rose from 76.0 to 83.3 using both tokens, illustrating the general efficacy of these mechanisms.

**MeanShift Shows Superior Performance and Stable Effects.** Among the evaluated methods, MeanShift performs most effectively and demonstrates more stable effects across different models, aligning with recent text-only steering findings (Wu et al., 2025). MeanShift's superiority and stability stem from its robustness: while SAE relies on learned sparse representations that may suffer from overfitting or incomplete concept capture, and probing operates in lower-dimensional space with sensitivity to specific projections, MeanShift operates on full-dimensional representations using distributional properties. This gives it more deterministic and stable effects across different models.

**Prompting Barely Steers.** Table 1 indicates that prompting is often less effective than targeted interventions and sometimes even deleterious. This deviates from text-only observations (Wu et al., 2025), reflecting MLLMs' challenges in following fine-grained visual reasoning instructions. Unlike text-only models that reliably execute linguistic guidance, multimodal models may struggle with translating textual prompts into enhanced visual understanding, making prompting less effective.

**Steering More Impactful for Spatial Relationships.** Interventions yield more substantial accuracy improvements in the "Spatial Relationship" task than in "Counting". For instance, as shown in Table 1, with both tokens and MeanShift, PaliGemma2-3B gained +7.3 for relationships but only +2.7 for counting. This disparity may stem from spatial relationships being more directly influenced by highlighting salient object features and positions, while counting might demand a more holistic scene interpretation, less directly aided by these specific steering methods.

**Smaller Models Show Better Steering Responsiveness.** Steering effectiveness increases with smaller model size, with the 3B model showing consistently larger improvements than the 10B model. This suggests that smaller models have more malleable internal representations, making them more receptive to steering interventions. For instance, PaliGemma2-3B demonstrates high responsiveness across all intervention types, while the 10B model shows reduced sensitivity to steering.

**Intervention Transfers Across Tasks.** As shown in Figure 5, intervention using a feature $\mathcal{T}$ some-times transfer effectively to different tasks $\mathcal{T}'$. For instance, enhancing attribute and entity recognition improves spatial relationship performance, suggesting that accurate object identification helps spatial reasoning. This cross-task transfer reflects the interconnected nature of visual understanding, where strengthening one capability can have cascading benefits for related reasoning processes.

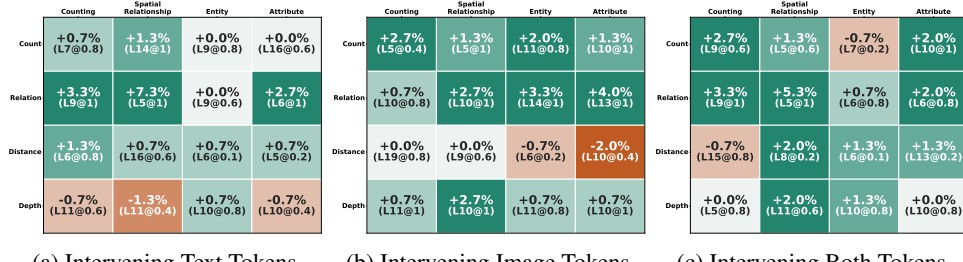

(a) Intervening Text Tokens      (b) Intervening Image Tokens      (c) Intervening Both Tokens

Figure 5: Performance improvements on CV-Bench tasks when steering PaliGemma2-3B with MeanShift vectors. Each cell shows the percentage improvement in accuracy relative to the baseline. Rows represent different CV-Bench tasks, while columns represent different feature vectors used for steering. Text below improvements indicates the optimal layer number and intervention strength.

# 6 STEERING IMPROVEMENTS GENERALIZE OUT-OF-DISTRIBUTION

We now examine the ability of textual steering methods for MLLMs to generalize out-of-distribution, *i.e.*, to datasets on which the steering method's hyperparameters $(\ell, \alpha)$ have not been tuned.

## 6.1 SETUP

**Datasets.** We first examine the transferability of textual steering on five datasets specifically designed to benchmark isolated visual reasoning capabilities: What'sUp-A, What'sUp-B, BLINK Object Localization, CLEVR, and Super-CLEVR. What'sUp-A contains 408 images of pairs of household objects arranged in clear spatial relations of {"on", "under", "left", and "right"}, while What'sUp-B similarly contains 412 images with objects in the image closer in size (Kamath et al., 2023). The BLINK Object Localization category contains 122 questions related to bounding boxes for large objects (Fu et al., 2024b). Finally, we sampled 500 datapoints from CLEVR (Johnson et al., 2017) and 200 datapoints from Super-CLEVR (Li et al., 2023b) to evaluate the OOD accuracy of textual steering in counting.

**Steering Vector Hyperparameter Selection.** We examine the previous three steering methodologies—SAE, MeanShift, and Probe—with a single choice of layer $\ell$ and scale factor $\alpha$ chosen independently of the test dataset. Specifically, for each test dataset, we select the $(\ell, \alpha)$ pair that performed best on the corresponding CV-Bench task category (*e.g.*, "Relation" for the What'sUp datasets and Blink Object Localization focusing on spatial relationships, and "Count" for CLEVR and Super-CLEVR).

We emphasize that the steering methods' hyperparameters are *not* tuned to the datasets considered in this section, making this a true test of out-of-distribution generalization. Similarly, our prompting baseline uses the exact prompt prefix that performed best on the associated CV-Bench tasks. The only adaptation made was the use of a small validation subset (50 datapoints for What'sUp and CLEVR, 25 datapoints for BLINK Object Localization and Super-CLEVR) to determine the most effective token type for intervention (image, text, or both) before evaluating on the remaining data.

## 6.2 RESULTS

**Steering Remains Broadly Effective.** Table 2 demonstrates that textual interventions are effective across all 5 datasets considered, attaining an average improvement over all models and datasets of at least +3.9 for all vector-based steering methods, demonstrating the strong OOD generalization of steering. Prompting averaged a +0.8 improvement and worsened model performance in 5 cases, suggesting that it may be less effective for MLLMs than for text-only LLMs (Wu et al., 2025).

**Validation Against Linguistic Bias.** The improved performance of steering on the What'sUp datasets provides evidence that our steering enhances genuine visual understanding rather than exploiting linguistic patterns. These datasets contain controlled image groups where identical objects are arranged in systematically varied spatial relationships (e.g., an apple positioned left, right, above, or

| DATASET | VISUAL CONCEPT | MODEL | BASELINE | INTERVENTION METHOD | | | |
|---|---|---|---|---|---|---|---|
| | | | | PROMPTING | SAE | PROBE | MEANSHIFT |
| What'sUp-A | Spatial Relation | PaliGemma2-3B | 62.7 | 65.8 (+3.1)⋆ | 71.8 (+9.1)⋆ | 78.5 (+15.8)⋆ | 75.4 (+12.7)⋆ |
| | | PaliGemma2-10B | 68.5 | 63.3 (−5.2) | 80.1 (+11.6)⋆ | 71.6 (+3.1)⋆ | 74.9 (+6.4)⋆ |
| | | Idefics3-8B-Llama3 | 62.2 | 61.9 (−0.4) | 64.1 (+1.9) | 62.2 (+0.0) | 61.9 (−0.3) |
| | | AVERAGE IMPROVEMENT | – | -0.8 | **+7.6** | +6.3 | +6.3 |
| What'sUp-B | Spatial Relation | PaliGemma2-3B | 60.6 | 56.7 (−3.9) | 58.9 (−1.7) | 57.5 (−3.1) | 60.3 (−0.3) |
| | | PaliGemma2-10B | 81.8 | 77.8 (−3.0) | 82.4 (+0.6) | 82.1 (+0.3) | 82.1 (+0.3) |
| | | Idefics3-8B-Llama3 | 52.0 | 57.3 (+5.3)⋆ | 56.2 (+4.2)⋆ | 57.0 (+5.0)⋆ | 63.4 (+11.5)⋆ |
| | | AVERAGE IMPROVEMENT | – | -0.5 | +1.0 | +0.8 | **+3.8** |
| BLINK Object Localization | Spatial Relation | PaliGemma2-3B | 41.2 | 41.2 (+0.0) | 43.3 (+2.1) | 42.3 (+1.0) | 44.3 (+3.1)⋆ |
| | | PaliGemma2-10B | 51.6 | 52.6 (+1.0) | 54.6 (+3.1) | 53.6 (+2.1) | 57.7 (+6.2)⋆ |
| | | Idefics3-8B-Llama3 | 53.6 | 53.6 (+0.0) | 56.7 (+3.1)⋆ | 53.6 (+0.0) | 55.7 (+2.1) |
| | | AVERAGE IMPROVEMENT | – | +0.3 | +2.8 | +1.0 | **+3.8** |
| CLEVR | Count | PaliGemma2-3B | 52.4 | 53.6 (+1.2) | 70.7 (+18.2)⋆ | 56.4 (+4.0)⋆ | 67.1 (+14.7)⋆ |
| | | PaliGemma2-10B | 70.7 | 72.4 (+1.7) | 74.9 (+4.2)⋆ | 71.6 (+0.9) | 80.4 (+9.8)⋆ |
| | | Idefics3-8B-Llama3 | 59.8 | 60.2 (+0.4) | 88.0 (+28.2)⋆ | 84.4 (+24.7)⋆ | 94.0 (+34.2)⋆ |
| | | AVERAGE IMPROVEMENT | – | +1.1 | +16.9 | +9.9 | **+19.6** |
| Super-CLEVR | Count | PaliGemma2-3B | 26.9 | 30.3 (+3.4) | 32.0 (+5.1)⋆ | 30.3 (+3.4) | 33.1 (+6.3)⋆ |
| | | PaliGemma2-10B | 40.0 | 48.5 (+8.5)⋆ | 40.6 (+0.6) | 40.0 (+0.0) | 44.6 (+4.6)⋆ |
| | | Idefics3-8B-Llama3 | 66.5 | 65.7 (−0.8) | 66.5 (+0.0) | 67.5 (+1.0) | 68.5 (+2.0)⋆ |
| | | AVERAGE IMPROVEMENT | – | +3.7 | +1.9 | +1.5 | **+4.3** |
| | AVERAGE IMPROVEMENT | | – | +0.8 | +6.0 | +3.9 | **+7.6** |

Table 2: Performance of textual steering on out-of-distribution datasets. Stars (⋆) denote statistically significant improvements ($p < 0.05$).

below the same plate). If we were merely exploiting textual patterns, we would expect biased outputs regardless of visual content, rather than the observed accurate tracking of true spatial relationships.

**Superior OOD Performance on Focused Tasks.** Remarkably, out-of-distribution performance often surpasses in-distribution results on CV-Bench, particularly on datasets requiring "pure" reasoning abilities. For example, CLEVR, which isolates counting skills using simple geometric objects without requiring complex object recognition, shows pronounced gains (+19.6 average). In contrast, CV-Bench Count and Super-CLEVR demand broader compositional understanding and object recognition beyond the targeted abilities, resulting in more moderate improvements. This pattern suggests our steering precisely targets the intended cognitive capabilities.

**MeanShift Demonstrates Consistent Superiority.** Across all experimental conditions, MeanShift consistently outperforms other extraction methods, achieving the highest average improvement of +7.6 compared to +6.0 for SAE and +3.9 for Linear Probing. This mirrors results from CV-Bench and AxBench (Wu et al., 2025), demonstrating MeanShift's consistent superiority across different domains and modalities.

## 6.3 RESULTS ON REAL-WORLD TASKS

The datasets evaluated in the previous subsection were specifically designed to benchmark isolated visual reasoning capabilities—spatial relationships and counting—making them ideal for controlled evaluation of our steering methods. To examine broader practical applicability, we further evaluated our cross-modal steering approach on real-world multimodal tasks that MLLMs encounter in practical applications, including: general visual question answering (VQAv2 (Goyal et al., 2017)), open-ended image captioning (COCO Captions (Chen et al., 2015)), document understanding (DocVQA (Mathew et al., 2021)), chart understanding (ChartQA (Masry et al., 2022)), and table reasoning (VTab-Fact (Kim et al., 2024)). We applied the most conceptually related steering vectors to their corresponding tasks, using the same experimental protocol as Section 6.1. Both the experimental details and complete results are provided in Appendix C.

Our steering methods demonstrate consistent effectiveness across these diverse domains, with Mean-Shift achieving improvements in 15 out of 18 model-task combinations and 7 statistically significant gains. While improvement magnitudes are smaller than those observed on our primary OOD datasets, this is expected since these complex tasks depend less exclusively on core spatial relation and counting skills that our steering vectors specifically target. Despite these differences, the consistent positive

| Task | Data Type | LoRA Performance | | | Average Improvement |
|------|-----------|-----------------|---|---|---------------------|
| | | PaliGemma-3B | PaliGemma-10B | Idefics-8B | |
| CVBench Relation | In-dist | 91.3 (+15.3)$^\star$ | 91.3 (+12.0)$^\star$ | 88.0 (+12.7)$^\star$ | +13.3 |
| CVBench Count | In-dist | 67.3 (+8.0)$^\star$ | 72.0 (+8.7)$^\star$ | 67.3 (+8.0)$^\star$ | +8.2 |
| Average In-Distribution | | +11.7 | +10.4 | +10.4 | **+10.8** |
| What'sUp-A | OOD | 67.7 (+5.0)$^\star$ | 69.3 (+0.8) | 61.6 (−0.6) | +1.7 |
| What'sUp-B | OOD | 58.4 (−2.2) | 86.0 (+4.2)$^\star$ | 58.1 (+6.1)$^\star$ | +2.7 |
| BLINK Object | OOD | 42.3 (+1.1) | 49.5 (−2.1) | 52.6 (+1.0) | +0.0 |
| CLEVR | OOD | 54.2 (+1.8) | 68.7 (−2.0) | 66.7 (+6.9)$^\star$ | +2.2 |
| Super-CLEVR | OOD | 28.6 (+1.7) | 43.4 (+3.4) | 66.9 (+0.4) | +1.8 |
| Average Out-of-Distribution | | +1.3 | +1.2 | +2.8 | **+1.7** |

Table 3: Performance comparison between LoRA and baseline models across in-distribution and out-of-distribution tasks. Stars ($\star$) denote statistically significant improvements ($p < 0.05$).

impact—especially the statistically significant gains—strongly indicates that our steering approach effectively enhances visual reasoning capabilities across diverse applications.

## 7 STEERING VS. FINE-TUNING

Beyond our interpretable steering methods, fine-tuning represents another common approach for enhancing model performance on specific tasks. To provide context for our steering approach, we compare against Low-Rank Adaptation (LoRA) fine-tuning (Hu et al., 2022) on the same tasks. We trained LoRA adapters using the training dataset from our grid search with an 80:20 train-validation split with hyperparameters: rank $r \in \{1, 2, 4\}$, alpha $\alpha \in \{4, 8\}$, learning rate $\eta \in \{1 \times 10^{-5}, 5 \times 10^{-5}, 1 \times 10^{-4}\}$, epochs = 3, and dropout = 0.1. We applied LoRA to the query and value projection parameters at the same layers used in our grid search: layers 5-20 for `PaliGemma2-3B` and `Idefics3-8B`, and layers 15-30 for `PaliGemma2-10B`. For each model and task combination, we selected hyperparameters that achieved optimal validation performance.

Table 3 presents the performance comparison between LoRA fine-tuning and our baseline models across in-distribution and out-of-distribution tasks. LoRA demonstrates strong in-distribution performance with an average improvement of +10.8 on CV-Bench, but its effectiveness diminishes significantly on out-of-distribution datasets with only +1.7 average improvement. In contrast, our steering methods maintain consistent performance across diverse datasets, with MeanShift achieving +7.6 and SAE achieving +6.0 average out-of-distribution improvements, highlighting the superior generalization capabilities of steering. This performance differential reflects fundamental differences in their mechanisms: LoRA adapts models to specific task distributions, while steering enhances underlying cognitive abilities such as spatial reasoning that remain applicable across diverse contexts.

## 8 DISCUSSION

We examine the ability of multimodal large language models (MLLMs) to be steered using textual steering vectors from their text-only backbone. We find that vectors extracted from Sparse Autoencoders (SAEs), Mean Shift, and Linear Probing can all enhance MLLMs' visual reasoning across diverse tasks on CV-Bench, with Mean Shift demonstrating the strongest overall performance. Notably, steering vectors with hyperparameters optimized on CV-Bench generalize to other out-of-distribution datasets with superior performance compared to LoRA fine-tuning or prompt tuning, underscoring text-driven steering as a powerful and efficient medium for enhancing visual reasoning in MLLMs. A primary limitation of our steering method is the reliance on the quality of extracted steering vectors. While existing vector extraction methods are widely used in the LLM interpretability community, the vectors they extract can be of poor quality and fail to adequately represent the target concepts, particularly for SAE and Linear Probing, leading to variable steering performance across different layers and models. Future work can focus on developing more robust extraction methods for text-only or cross-modal models to improve the reliability and consistency of steering vectors.

## ETHICS STATEMENT

We identify no significant ethical concerns. Our steering methods enhance visual reasoning on standard benchmarks without introducing inherently harmful capabilities. While these techniques could potentially be misused like any model modification approach, the risk is not greater than that of the underlying MLLMs. We encourage responsible use and consideration of societal impacts when deploying enhanced MLLMs.

## REPRODUCIBILITY STATEMENT

We provide comprehensive implementation details and have open-sourced our code on GitHub and uploaded it as supplementary material to OpenReview. Steering vector extraction methods are detailed in Section 4.2 and Algorithm 1. Hyperparameter grid search procedures and experimental protocols are described in Sections 5.1, 5, and 6. We use publicly available pre-trained SAEs (GemmaScope, LlamaScope), models (PaliGemma2, Idefics3), and datasets (CV-Bench, What'sUp, BLINK, CLEVR, Super-CLEVR, VQAv2, COCO Captions, DocVQA, ChartQA, VTabFact).

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

# APPENDIX

## A   STEERING VECTOR METHODOLOGY

### A.1   SPARSE AUTOENCODERS

We now provide further detail regarding the extraction of textual steering vectors for visual concepts using SAEs.

Recall that we consider four important taxonomies for image-related concepts: spatial relationship, counting, attribute, and entity. For each taxonomy, we sample $K$ sentences $\{s_1, \ldots, s_K\}$ containing these visual concepts. In practice, we set $K$ to 20. For each sentence $s_j$, we identify the anchor word for this visual concept as $w_j$, thus forming sentence-anchor pairs $(s_j, w_j)$. See table 4 for several examples.

Table 4: Sample sentence and anchor word pairs for various taxonomies.

| TAXONOMY | SENTENCE $s_j$ | ANCHOR WORD $w_j$ |
|---|---|---|
| Spatial Relationship | The cat is on the table
She put the book under the chair | on
under |
| Counting | There are three apples in the basket
The teacher counted five children | three
five |
| Attribute | The red car stopped at the light
She wore a beautiful dress | red
beautiful |
| Entity | The dog barked at the mailman
A tree fell during the storm | dog
tree |

Using our sentence-anchor pairs, we identify features with high activations on anchor words. Interestingly, as shown in Figure 6, we find that each visual concept activates only a limited number of SAE features, indicating a sparse encoding of these concepts. We then verify their relevance to the target visual concepts and average these relevant feature vectors to create a single steering vector for each visual concept at each layer.

We then use these sentence-anchor pairs to identify feature directions corresponding to the ideal visual concepts using Algorithm 1. We employ a two-stage procedure which, at the first stage, finds the top $n$ activated features for anchor words $w_j$ in sentences $s_j$. At the second stage, we use `o3-mini` (OpenAI, 2025) to verify that these features indeed align with the desired visual concept $\mathcal{C}$. To accomplish the procedure, we use pretrained SAEs with detailed explanations and top activations developed by the interpretability community, such as `GemmaScope` (Lieberum et al., 2024b) for `Gemma-2-2B` and `Gemma-2-9B` (Team, 2024b), and `LlamaScope` (He et al., 2024) for `Llama-3.1-8B` base model (at Meta, 2024). When we prompt `o3-mini` for verification, we craft

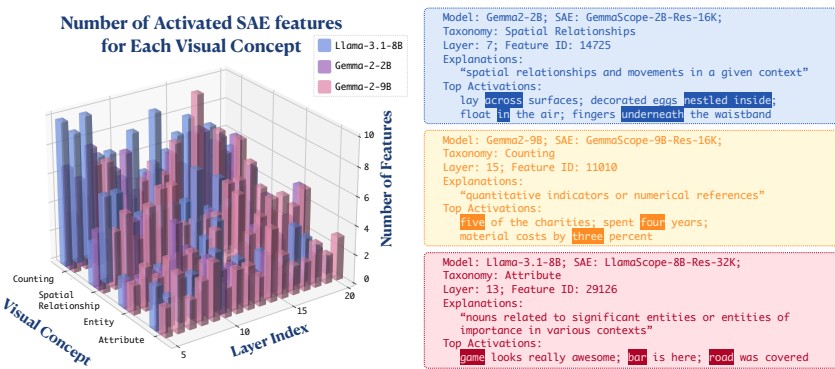

Figure 6: **Left**: Number of SAE features associated with each taxonomy (counting, spatial relationship, entity, and attribute) across the layers of Llama-3.1-8B, Gemma2-2B, and Gemma2-9B. Notably, SAE features for such visual concepts are sparse, numbering fewer than 10 across 16k total SAE features (Gemma2-2B/9B) or 32k features (Llama-3.1-8B). **Right**: Examples of features corresponding to visual concepts, identified by the layer whose activation space they inhabit and their (arbitrary) feature ID. The feature's explanation summarizes its semantic meaning, as evidenced by the tokens and contexts on which it attains the greatest activations.

prompts to include both the explanation for the candidate feature vector $v_i^{(\ell)}$, and sample top activated tokens (see figure 7 for the prompting template). We find that `o3-mini` can indeed filter out features unrelated to the desired visual concepts.

---

**Algorithm 1** Find Textual Representations for Visual Concepts using SAEs

---

**Require:** Desired visual concepts $\mathcal{C}$. Layer index $\ell$.
**Require:** Sentence and anchor word pairs $\{(s_1, w_1), \cdots, (s_K, w_K)\}$.
**Require:** Pretrained SAEs at layer $\ell$.
    ▷ Find top activations and their corresponding SAE feature vectors.
    $\mathcal{V}_0 = \{\}$
    **for** each $(s_j, w_j)$ **do**
        $\{\alpha_i^{(\ell)}(w_j), v_i^{(\ell)}\} \leftarrow$ Pass $s_j$ into the pretrained SAE
        $\{v_{i_1}^{(\ell)}, \cdots v_{i_n}^{(\ell)}\} \leftarrow \text{Top}_n\{\alpha_i^{(\ell)}(w_j), v_i^{(\ell)}\}$ ranked by activation strength $\alpha_i^{(\ell)}(w_j)$
        $\mathcal{V}_0 \leftarrow \mathcal{V}_0 \cup \{v_{i_1}^{(\ell)}, \cdots v_{i_n}^{(\ell)}\}$
    **end for**
    ▷ Filter out noisy SAE feature vectors.
    $\mathcal{V} = \{\}$
    **for** each $v_i^{(\ell)} \in \mathcal{V}_0$ **do**
        Find the explanation $e$ and top activated tokens $\{t_1, \cdots, t_p\}$ for $v_i^{(\ell)}$
        **if** `o3-mini`(VerificationPrompt, $e$, $\{t_1, \cdots, t_p\}$, $\mathcal{C}$) is True **then**
            $\mathcal{V} \leftarrow \mathcal{V} \cup \{v_i^{(\ell)}\}$
        **end if**
    **end for**
    ▷ Aggregate SAE vectors to one steering vector.
    $v^{(\ell)} = \frac{1}{|\mathcal{V}|} \sum_{u \in \mathcal{V}} u$
    **return** $v^{(\ell)}$

---

## A.2 PROMPTING

We now elaborate upon our generation of prompts for eliciting taxonomy-specific visual reasoning in MLLMs. As described in Section 4.2, we generate a total of 96 candidate prompts for each taxonomy $\mathcal{T}$. To do so, we use template shown in figure 8. Here, we set the `num instructions` to 6 and `word`

```
FEATURE ALIGNMENT VERIFICATION

Task:  Determine if a neural network's sparse autoencoder (SAE)
feature aligns with the taxonomy "{taxonomy}".

Taxonomy Definition:  {taxonomy_definition}

Feature Information:
1.  Feature's explanation:  {feature_explanation}
2.  Top activation examples (tokens wrapped in <top>...</top> have the
highest activation values and are the most important to focus on):
    1.  {activation_example_1}
    2.  {activation_example_2}
    3.  {activation_example_3}
    4.  {activation_example_4}
    5.  {activation_example_5}

Examples of features that DO align with the {taxonomy} taxonomy
(notice how the key words are highlighted with <top>...</top> tags):
Example 1:
- Explanation:  {explanation_1}
- Activations:  {activations_1}
Example 2:
- Explanation:  {explanation_2}
- Activations:  {activations_2}

When making your decision, you should follow these rules:
1.  First pay attention to the feature's explanation.
2.  If you cannot decide, you should then pay special attention to
the tokens highlighted with <top>...</top> tags, as these are the most
highly activated tokens and strongest indicators of what the feature
detects.
3.  Also consider the diversity of the activation examples provided.
If one feature only activates one particular word, it may not be as
aligned as a feature that activates on a variety of words.

Based on the feature's explanation and the highlighted tokens in the
activation examples, does this feature specifically detect or respond
to {taxonomy_definition}?  Your answer should start with YES or NO,
then provide a brief reason.  Do not start with any other words or
phrases such as 'answer'.
```

Figure 7: Prompt template for querying GPT-o3-mini to verify whether a given feature is related to a visual taxonomy. For each taxonomy, the template employs a brief definition of the taxonomy, two example features that align with each taxonomy (for few-shot learning), and the top five activations of the feature in question.

count $\in \{5, 10, 15, 20, 25, 30, 35, 40, 45, 50, 55, 60, 65, 70, 75, 80\}$, resulting in total $6 \times 16 = 96$ steering prompts.

---

**STEERING PROMPT GENERATION**

**System prompt:** You are an expert at creating concise, clear instructions for Multimodal Large Language Models (MLLM).

Your task:
- Generate {num_instructions} different instruction(5) that will make the Model focus on {concept} when answering questions about images
- Each instruction must be within {word_count} words
- Instructions should be direct and actionable, focusing specifically on how to emphasize {concept}

IMPORTANT FORMAT REQUIREMENTS:
- Begin each instruction with "INSTRUCTION:" followed by the instruction text
- Put each instruction on its own line
- Do not include any numbering, bullets, or other text beyond the requested instructions
- Do not include any explanations, introductions, or conclusions

Example format for 2 instructions:
INSTRUCTION: First instruction text here within word limit.
INSTRUCTION: Second instruction text here within word limit.

**User prompt:** Create {num_instructions} instruction(s) about {concept} using {word_count} words or fewer each.

Figure 8: System and user prompt template for generating MLLM prompts.

# B  ADDITIONAL COLOR PERCEPTION INTERVENTION EXAMPLES

To further demonstrate the effectiveness of textual steering vectors in modifying visual understanding within MLLMs, we present additional color perception intervention examples using the same methodology described in §3.

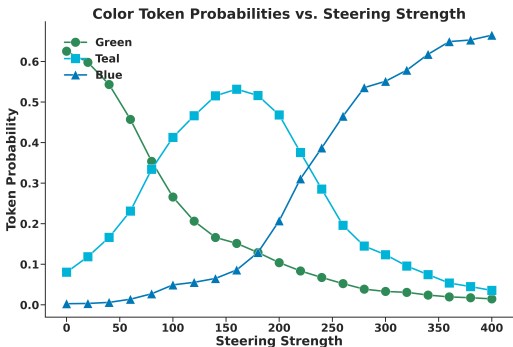

(a) Steering a green image toward blue perception. As the scale factor increases, the model's interpretation shifts from green to teal, and ultimately to blue.

(b) Steering a purple image toward red perception. The intervention gradually shifts the model's color association from purple to pink, and finally to red.

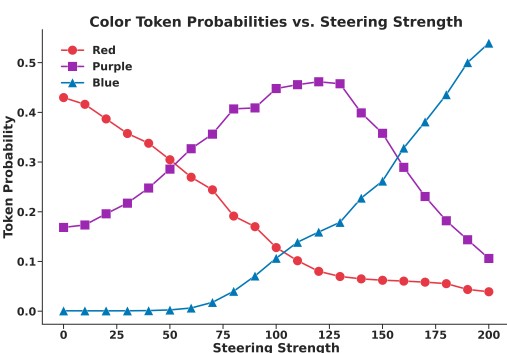
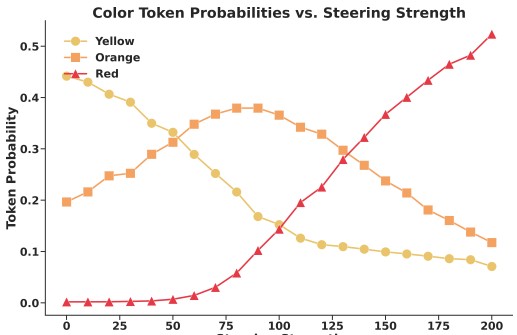

(c) Steering an red image toward blue perception. The intervention causes a gradual shift from red to purple, and ultimately to blue.

(d) Another example of steering a yellow image toward red perception, using a different steering vector from layer 18 of `PaliGemma2-10B`. As the scale factor increases, the model's interpretation transitions from yellow to orange, and finally to red.

Figure 9: Additional color perception intervention examples. In each case, we apply the normalized textual steering vector for the target color to the image tokens with increasing scale factors. The steering vectors are extracted from and applies to one selected layer from layer 17 to 20 in `PaliGemma2-10B`. The plots show token probability shifts, demonstrating how textual steering vectors can systematically modify the model's visual perception.

These additional examples further support our findings in §3. In each case, we see a clear progression of perception as the steering strength increases, with intermediate colors appearing during the transition. This confirms that textual steering vectors can produce predictable and continuous modifications to visual understanding.

Notably, all these interventions were performed using steering vectors derived solely from text data, yet they effectively modulate multimodal understanding. This provides additional evidence for our hypothesis that MLLMs develop unified cross-modal representations that can be manipulated through textual steering.

| TASK | VISUAL CONCEPT | MODEL | INTERVENTION METHOD | | | | |
|---|---|---|---|---|---|---|---|
| | | | BASELINE | PROMPTING | SAE | PROBE | MEANSHIFT |
| VQAv2 | Spatial Relations | `PaliGemma2-3B` | 86.8 | 87.0 (+0.2) | 88.2 (+2.4) | 87.1 (+0.3) | 89.3 (+3.5)★ |
| | | `PaliGemma2-10B` | 88.2 | 86.8 (−1.4) | 87.4 (−0.8) | 86.9 (−1.3) | 88.9 (+0.7) |
| | | `Idefics3-8B` | 76.7 | 76.7 (+0.0) | 78.1 (+1.4) | 77.8 (+1.1) | 74.6 (−2.1) |
| | | AVERAGE IMPROVEMENT | – | -0.4 | **+1.0** | +0.0 | +0.7 |
| COCO Captions | Spatial Relations | `PaliGemma2-3B` | 147.9 | 144.4 (−3.5) | 151.2 (+3.3)★ | 151.0 (+3.1)★ | 152.5 (+4.6)★ |
| | | `PaliGemma2-10B` | 155.8 | 141.3 (−14.5) | 160.0 (+4.2)★ | 161.1 (+5.3)★ | 158.4 (+2.6) |
| | | `Idefics3-8B` | 70.0 | 70.3 (+0.3) | 71.2 (+1.2) | 70.9 (+0.9) | 69.6 (−0.5) |
| | | AVERAGE IMPROVEMENT | – | -5.9 | +2.9 | **+3.1** | +2.2 |
| DocVQA Layout | Spatial Relations | `PaliGemma2-3B` | 79.4 | 81.4 (+2.0) | 84.8 (+5.4)★ | 81.0 (+1.6) | 85.4 (+6.0)★ |
| | | `PaliGemma2-10B` | 81.3 | 82.5 (+1.2) | 83.8 (+2.5) | 83.9 (+2.6)★ | 83.8 (+2.5)★ |
| | | `Idefics3-8B` | 88.5 | 86.3 (−2.2) | 89.6 (+1.1) | 88.2 (−0.3) | 89.7 (+1.3) |
| | | AVERAGE IMPROVEMENT | – | +0.3 | +3.0 | +1.3 | **+3.3** |
| DocVQA Number | Counting | `PaliGemma2-3B` | 76.1 | 76.2 (+0.1) | 75.8 (−0.3) | 76.3 (+0.2) | 76.5 (+0.4) |
| | | `PaliGemma2-10B` | 77.7 | 75.8 (−1.9) | 77.6 (−0.1) | 79.4 (+1.7) | 76.9 (−0.8) |
| | | `Idefics3-8B` | 86.8 | 84.5 (−2.3) | 87.3 (+0.5) | 86.9 (+0.1) | 89.8 (+3.0) |
| | | AVERAGE IMPROVEMENT | – | -1.4 | +0.0 | +0.7 | **+0.9** |
| ChartQA | Counting | `PaliGemma2-3B` | 46.4 | 45.4 (−1.0) | 46.6 (+0.2) | 47.4 (+1.0) | 48.0 (+1.6) |
| | | `PaliGemma2-10B` | 51.8 | 53.2 (+1.4) | 53.8 (+2.0) | 53.4 (+1.6) | 54.4 (+2.6)★ |
| | | `Idefics3-8B` | 68.2 | 67.4 (−0.8) | 71.0 (+2.8)★ | 67.2 (−1.0) | 72.6 (+4.4)★ |
| | | AVERAGE IMPROVEMENT | – | -0.1 | +1.7 | +0.5 | **+2.9** |
| VTabFact | Counting | `PaliGemma2-3B` | 56.5 | 54.5 (−2.0) | 58.0 (+1.5) | 56.0 (−0.5) | 60.5 (+4.0)★ |
| | | `PaliGemma2-10B` | 57.0 | 58.5 (+1.5) | 58.5 (+1.5) | 59.0 (+2.0) | 58.5 (+1.5) |
| | | `Idefics3-8B` | 70.0 | 71.0 (+1.0) | 75.5 (+5.5)★ | 71.0 (+1.0) | 73.5 (+3.5) |
| | | AVERAGE IMPROVEMENT | – | +0.2 | +2.8 | +0.8 | **+3.0** |

Table 5: Performance of textual steering methods on real-world multimodal tasks. Stars ($\star$) denote statistically significant improvements ($p < 0.05$).

## C  RESULTS ON REAL-WORLD TASKS

**Experimental Setup:** We evaluated our steering methods on six real-world multimodal tasks using 500 examples per dataset (200 for VTabFact due to dataset size limitations) for testing, and extra 50 examples for validation to determine optimal intervention token types. We applied counting steering vectors to numerical reasoning tasks and spatial relationship vectors to layout and captioning tasks.

**Task Details and Metrics:**

- **VQAv2:** General visual question answering task, evaluated using the official VQA Accuracy metric.
- **COCO Captions:** Open-ended image captioning task, evaluated using CIDEr-D metric.
- **DocVQA Layout:** Document QA task focusing on spatial layout and structure questions, evaluated using ANLS×100.
- **DocVQA Number:** Document QA task focusing on numerical information extraction, evaluated using ANLS×100.
- **ChartQA:** Chart interpretation and reasoning QA task, evaluated using the Relaxed Accuracy metric.
- **VTabFact:** Table reasoning multiple choice task, evaluated using accuracy.

Results demonstrate consistent effectiveness across diverse real-world applications, with MeanShift achieving improvements in 15 out of 18 model-task combinations. The smaller improvement magnitudes compared to capability-focused benchmarks reflect the multi-faceted nature of these tasks, which require comprehensive reasoning abilities beyond isolated spatial or counting skills.

# D  DATASET EVALUATION DETAILS

In this section, we explain in detail how we prompt and evaluate the model's performance across datasets and provide representative examples for each dataset. Each prompt consists of four components: `model prefix`, `task prefix`, `taxonomy prefix`, and `question`. The `model prefix` is the specific instruction token sequence required by different model families to perform certain tasks. For `PaliGemma2` models, we use `"answer en"` as the model prefix, indicating that the model should answer in English for visual question answering tasks. For COCO dataset specifically, we use `"caption en"`, indicating that it is a captioning task. For `Idefics3-8B-Llama3`, no model prefix is required, so this component remains empty. The `task prefix` provides task-specific instructions that constrain the format of the model's response. In multiple-choice questions, we use a task prefix such as `"Answer the multiple choice question by only responding with the letter of the correct answer."` for example. In CLEVR and Super-CLEVR counting questions, we use `"Answer the question by only responding the number."` The `taxonomy prefix` of each taxonomy is the prompt we sampled in Section A.2, and it is only non-empty for the `Prompt` method. The `question` component contains the original question format from the dataset. Below are examples illustrating our prompting approach for each dataset.

---

**CV-BENCH RELATION**

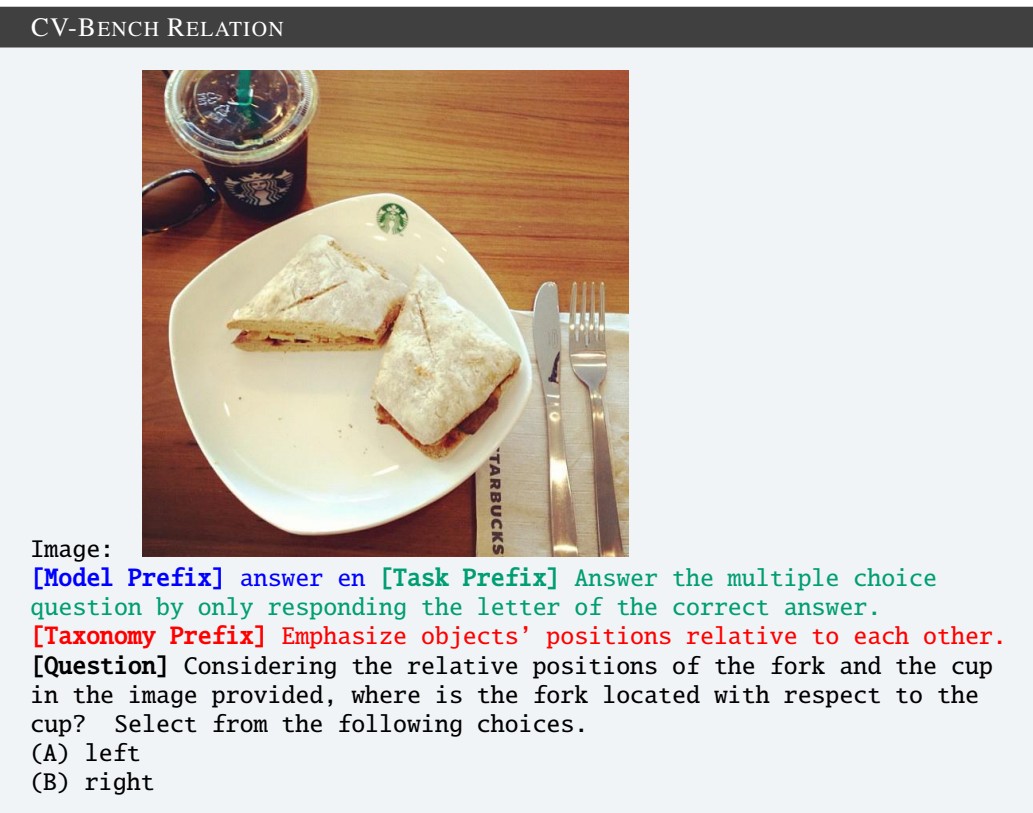

Image:
**[Model Prefix]** answer en **[Task Prefix]** Answer the multiple choice question by only responding the letter of the correct answer.
**[Taxonomy Prefix]** Emphasize objects' positions relative to each other.
**[Question]** Considering the relative positions of the fork and the cup in the image provided, where is the fork located with respect to the cup?  Select from the following choices.
(A) left
(B) right

---

Figure 10: Example prompt for the CV-Bench Relation dataset.

CV-BENCH COUNT

Image:
[Model Prefix] answer en [Task Prefix] Answer the multiple choice
question by only responding the letter of the correct answer.
[Taxonomy Prefix] Prioritize counting objects and quantifying
elements over other analysis.  [Question] Answer the multiple choice
question by only responding the letter of the correct answer.  How
many beds are in the image?  Select from the following choices.
(A) 0
(B) 2
(C) 1
(D) 3
(E) 4

Figure 11: Example prompt for the CV-Bench Count dataset.

WHAT'SUP-A

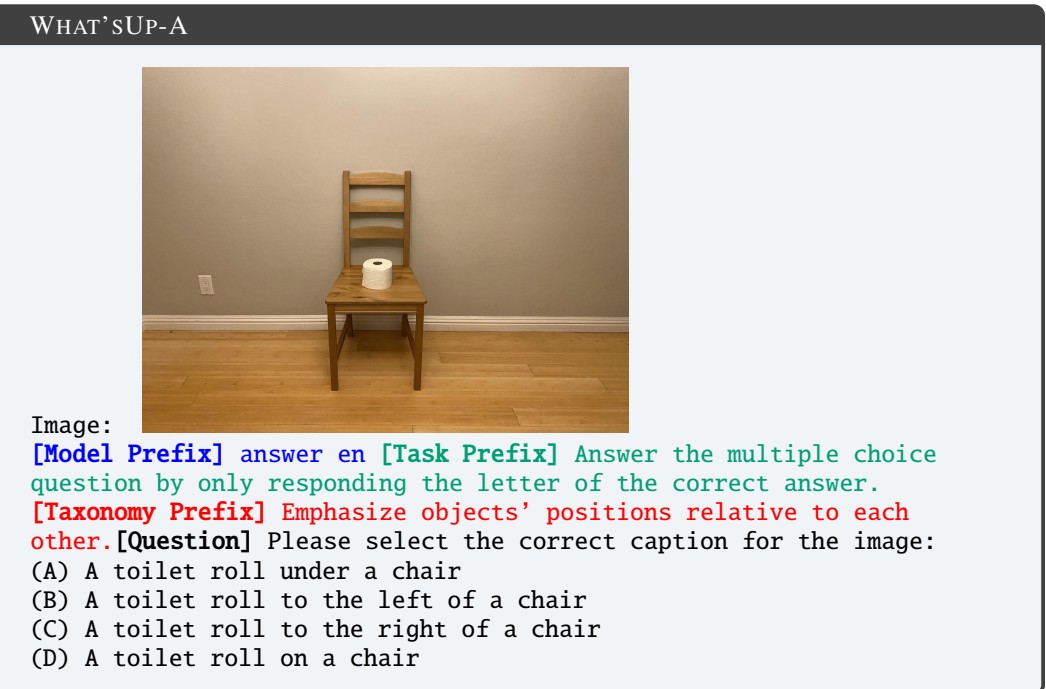

Image:
[Model Prefix] answer en [Task Prefix] Answer the multiple choice
question by only responding the letter of the correct answer.
[Taxonomy Prefix] Emphasize objects' positions relative to each
other.[Question] Please select the correct caption for the image:
(A) A toilet roll under a chair
(B) A toilet roll to the left of a chair
(C) A toilet roll to the right of a chair
(D) A toilet roll on a chair

Figure 12: Example prompt for the What'sUp-A dataset.

WHAT'SUP-B

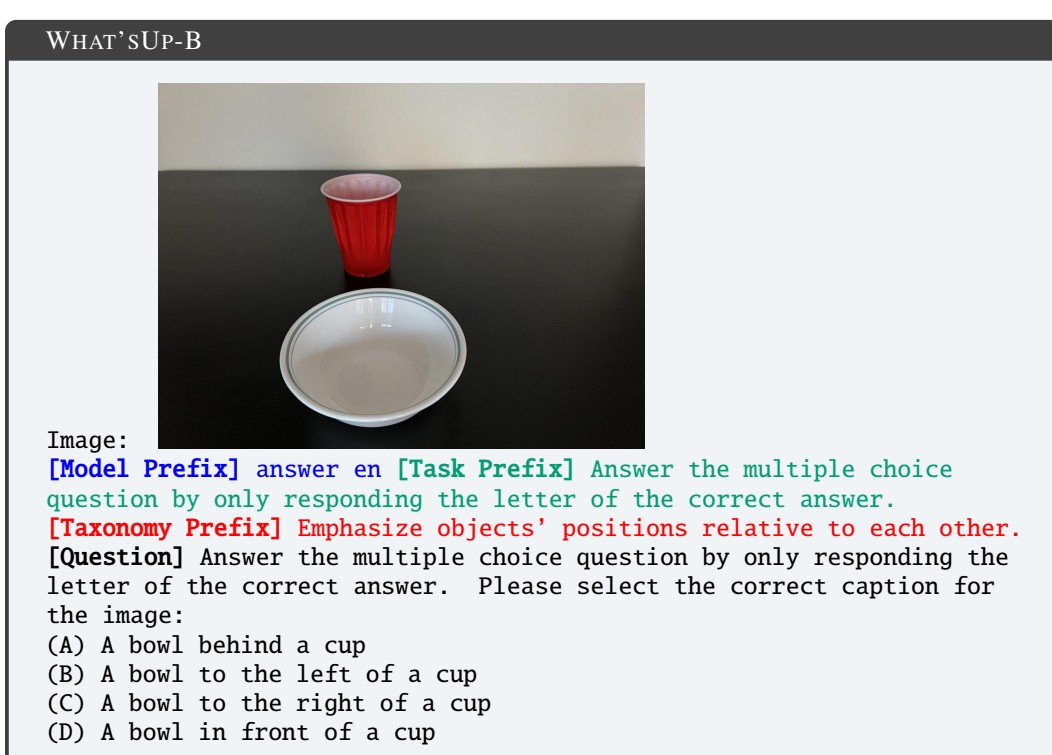

Image:
[Model Prefix] answer en [Task Prefix] Answer the multiple choice
question by only responding the letter of the correct answer.
[Taxonomy Prefix] Emphasize objects' positions relative to each other.
[Question] Answer the multiple choice question by only responding the
letter of the correct answer.  Please select the correct caption for
the image:
(A) A bowl behind a cup
(B) A bowl to the left of a cup
(C) A bowl to the right of a cup
(D) A bowl in front of a cup

Figure 13: Example prompt for the What'sUp-B dataset.

BLINK OBJECT LOCALIZATION

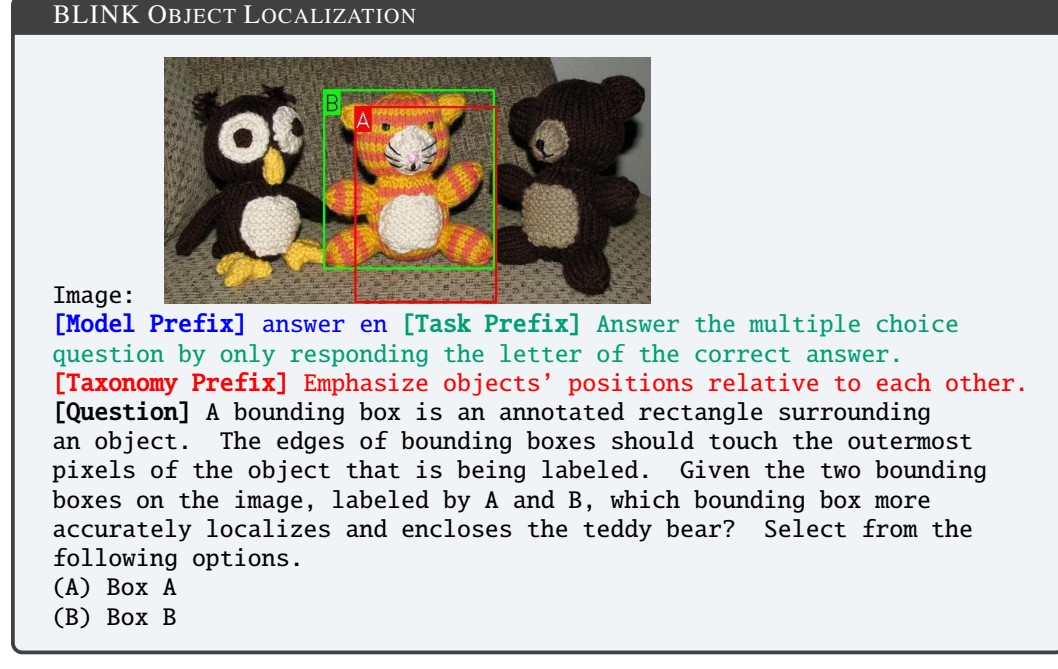

Image:
[Model Prefix] answer en [Task Prefix] Answer the multiple choice
question by only responding the letter of the correct answer.
[Taxonomy Prefix] Emphasize objects' positions relative to each other.
[Question] A bounding box is an annotated rectangle surrounding
an object.  The edges of bounding boxes should touch the outermost
pixels of the object that is being labeled.  Given the two bounding
boxes on the image, labeled by A and B, which bounding box more
accurately localizes and encloses the teddy bear?  Select from the
following options.
(A) Box A
(B) Box B

Figure 14: Example prompt for the BLINK Object Localization dataset.

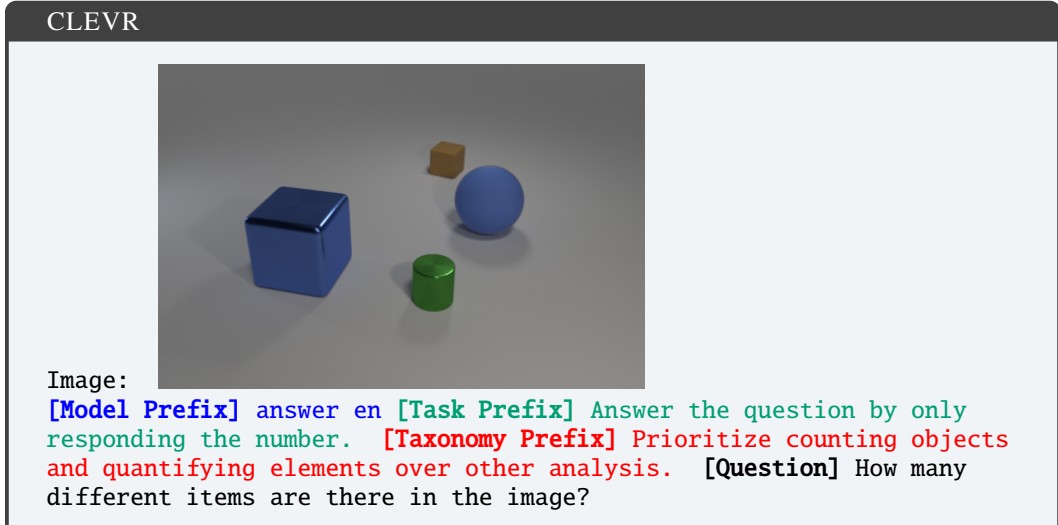

Image:
**[Model Prefix]** answer en **[Task Prefix]** Answer the question by only responding the number. **[Taxonomy Prefix]** Prioritize counting objects and quantifying elements over other analysis. **[Question]** How many different items are there in the image?

Figure 15: Example prompt for the CLEVR dataset.

SUPER-CLEVR

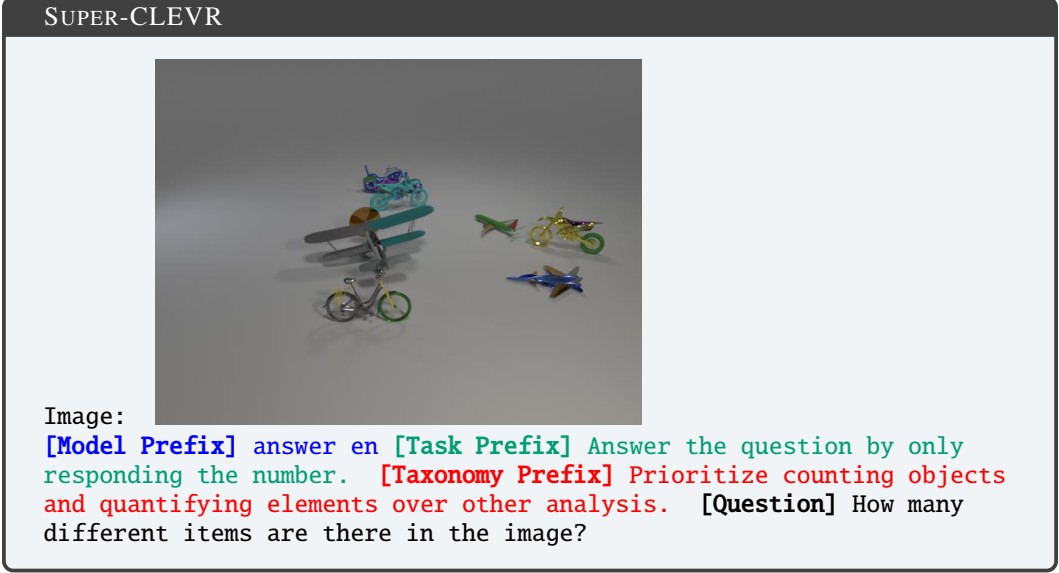

Image:
**[Model Prefix]** answer en **[Task Prefix]** Answer the question by only responding the number. **[Taxonomy Prefix]** Prioritize counting objects and quantifying elements over other analysis. **[Question]** How many different items are there in the image?

Figure 16: Example prompt for the Super-CLEVR dataset.

**VQAv2**

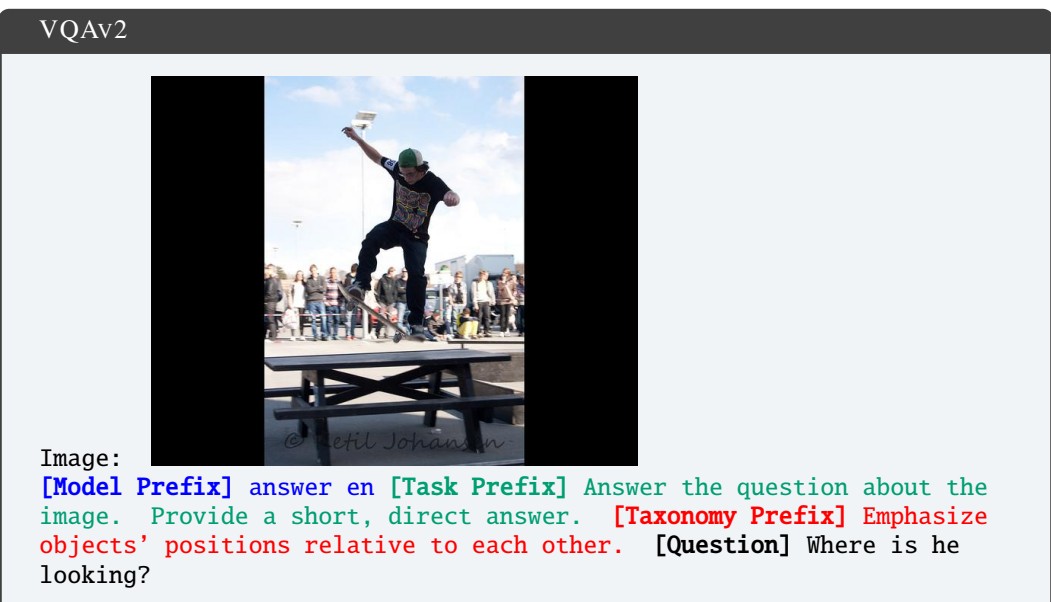

Image:
**[Model Prefix]** answer en **[Task Prefix]** Answer the question about the image. Provide a short, direct answer. **[Taxonomy Prefix]** Emphasize objects' positions relative to each other. **[Question]** Where is he looking?

Figure 17: Example prompt for the VQAv2 dataset.

**COCO**

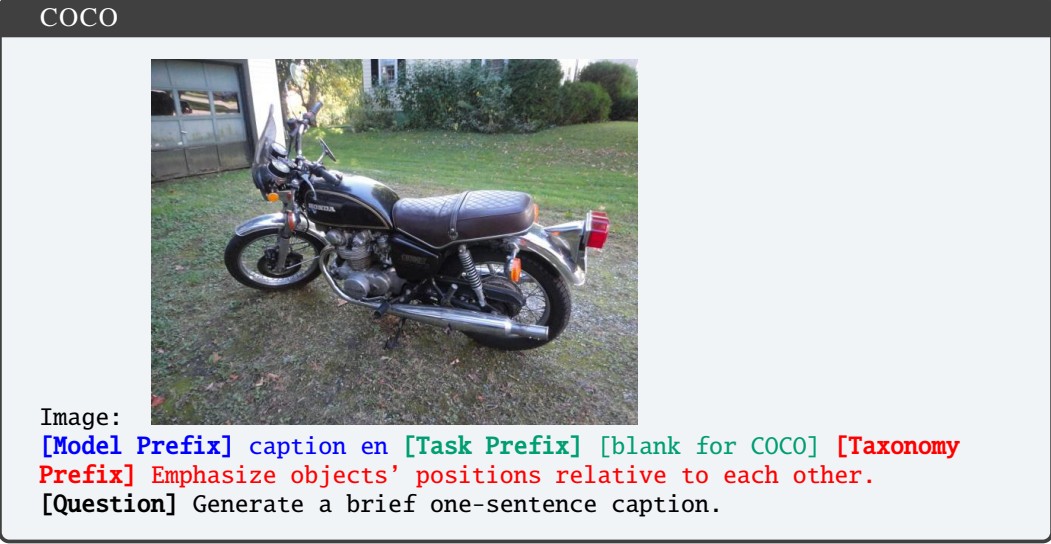

Image:
**[Model Prefix]** caption en **[Task Prefix]** [blank for COCO] **[Taxonomy Prefix]** Emphasize objects' positions relative to each other. **[Question]** Generate a brief one-sentence caption.

Figure 18: Example prompt for the COCO dataset.

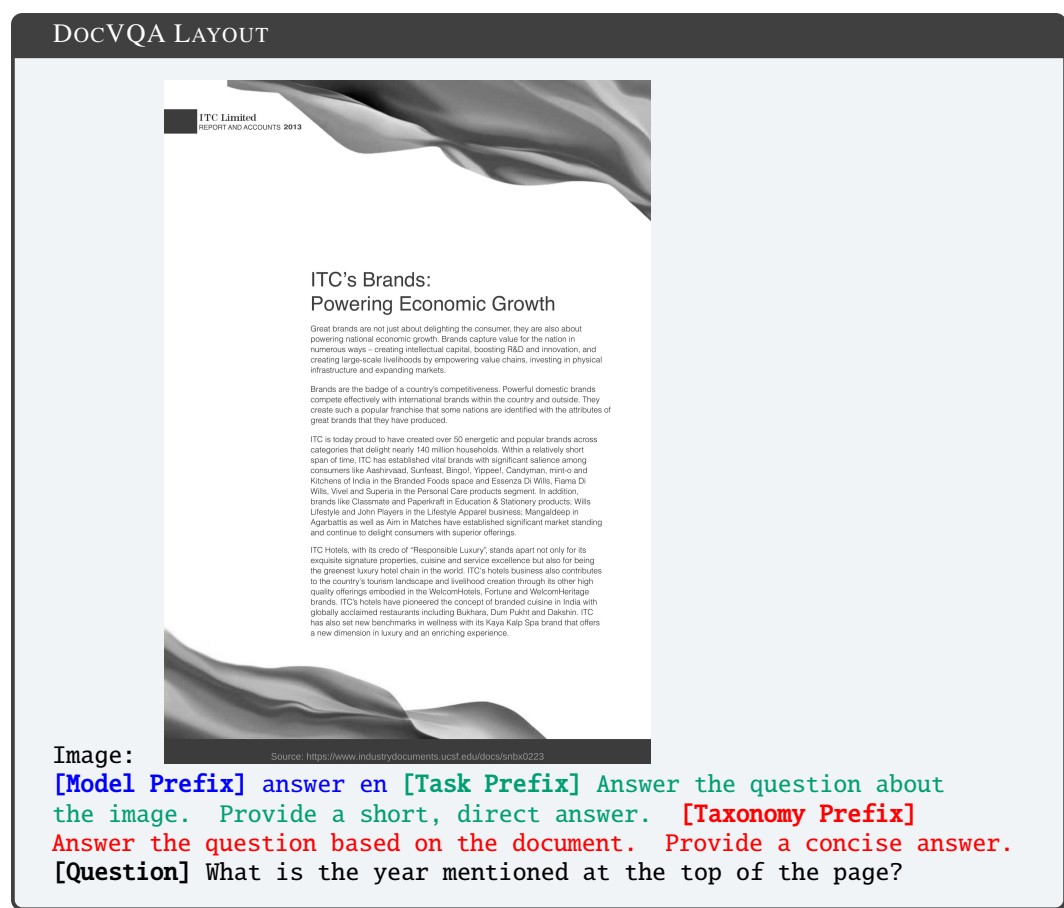

Figure 19: Example prompt for the DocVQA Layout dataset.

## DOCVQA NUMBER

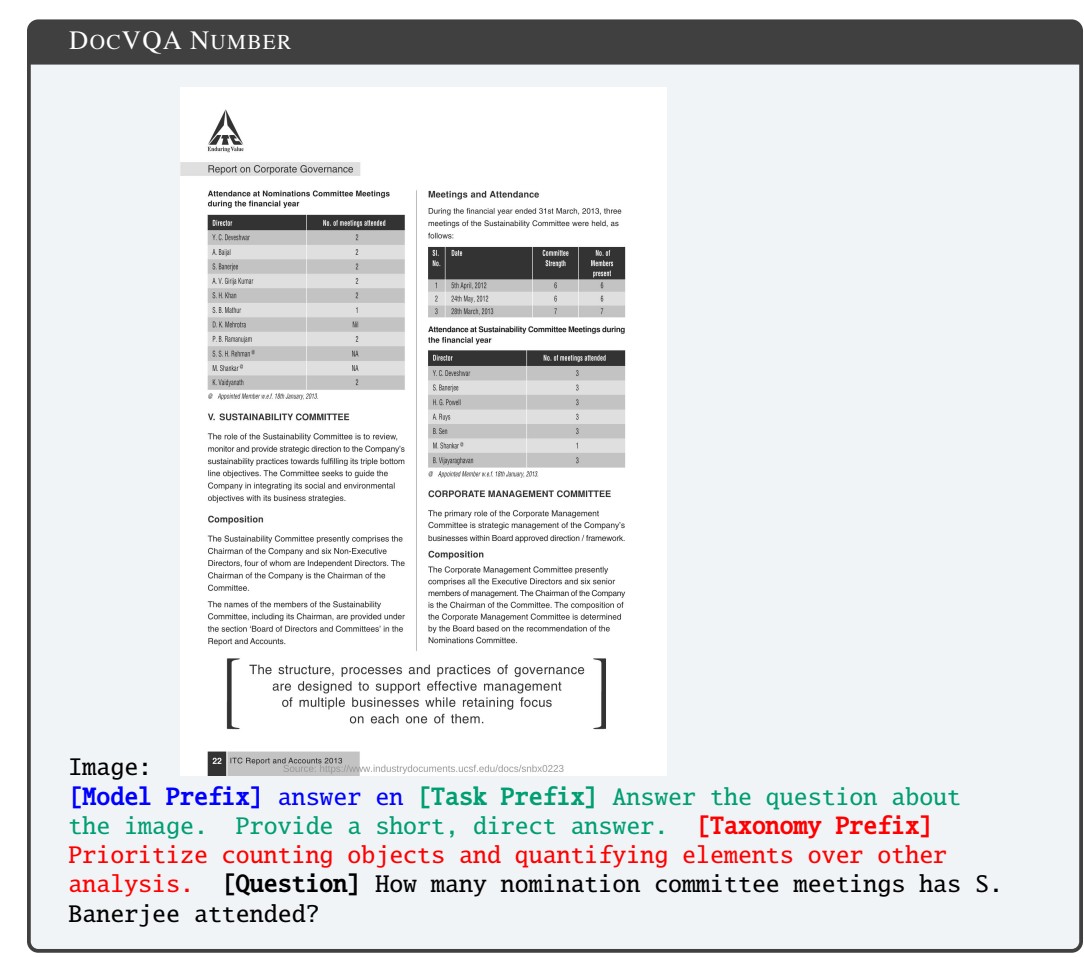

Image: [Model Prefix] answer en [Task Prefix] Answer the question about the image. Provide a short, direct answer. [Taxonomy Prefix] Prioritize counting objects and quantifying elements over other analysis. [Question] How many nomination committee meetings has S. Banerjee attended?

Figure 20: Example prompt for the DocVQA Number dataset.

CHARTQA

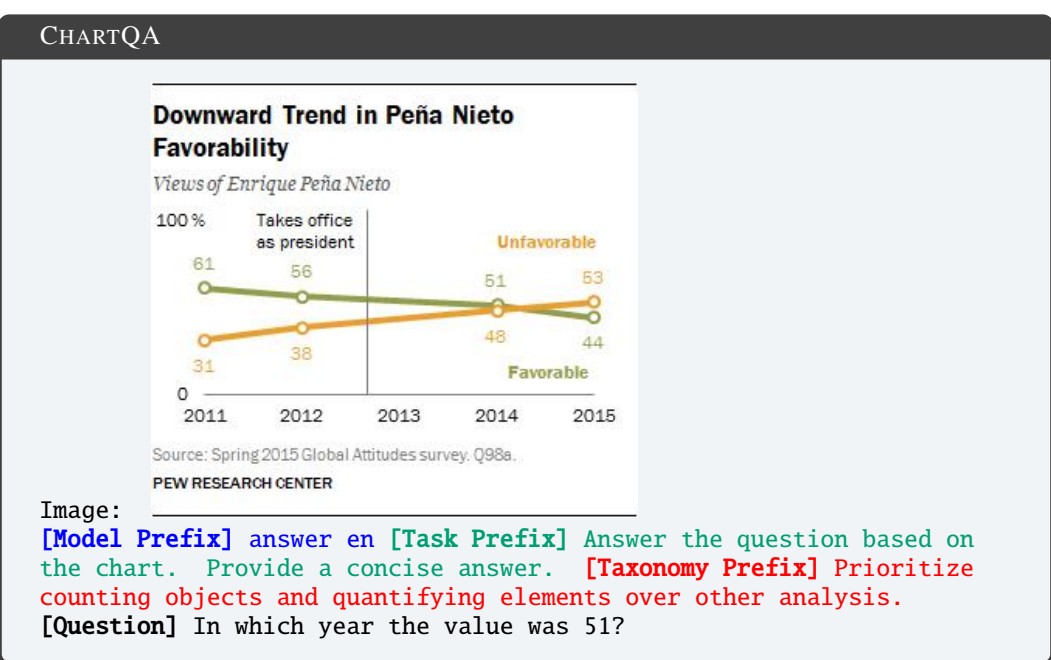

Image:
**[Model Prefix]** answer en **[Task Prefix]** Answer the question based on
the chart. Provide a concise answer. **[Taxonomy Prefix]** Prioritize
counting objects and quantifying elements over other analysis.
**[Question]** In which year the value was 51?

Figure 21: Example prompt for the ChartQA dataset.

VTABFACT

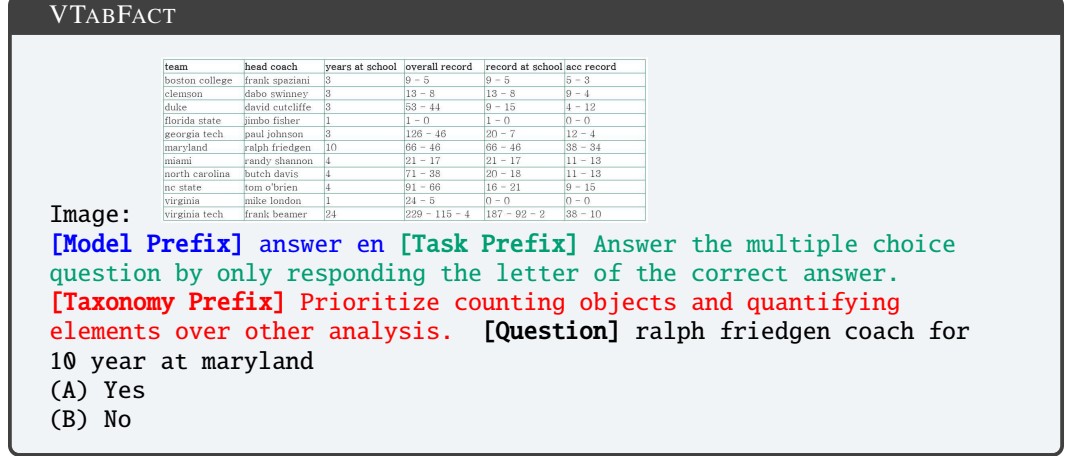

Image:
**[Model Prefix]** answer en **[Task Prefix]** Answer the multiple choice
question by only responding the letter of the correct answer.
**[Taxonomy Prefix]** Prioritize counting objects and quantifying
elements over other analysis. **[Question]** ralph friedgen coach for
10 year at maryland
(A) Yes
(B) No

Figure 22: Example prompt for the VtabFact dataset.

# E    LLM Usage Statement

LLMs were used for: (1) text polishing, (2) SAE feature verification using o3-mini (Section A.1), and (3) prompt generation using GPT-4o for baselines (Section A.2). These applications were limited to specific methodological components. Core research ideas, innovations, and conclusions are original author contributions.

