# OpenReview forum: "Textual Steering Vectors Can Improve Visual Understanding in Multimodal Large Language Models"
_ICLR.cc/2026/Conference — ICLR 2026 Conference Withdrawn Submission_

### Official Review · Reviewer_qPh5 · 2025-10-27

**Soundness:** 1
**Presentation:** 2
**Contribution:** 2
**Rating:** 2
**Confidence:** 4

**Summary:**

Textual steering vectors enable cross-modal transfer of interpretability tools, providing a lightweight, training-free mechanism to enhance visual reasoning in MLLMs.

**Strengths:**

The paper demonstrates a cross-modal transfer phenomenon that allows reusing textual interpretability tools to improve visual reasoning in multimodal models without any additional fine-tuning.

**Weaknesses:**

1. Limited Scope of Related Work on Cross-Modal Steering

The related work section is critically underdeveloped, failing to contextualize this work within the emerging and diverse landscape of cross-modal and multimodal steering. Specifically, the paper omits discussion of methods that directly learn steering from multimodal pairs, or approaches like the concurrent "LLaVA Steering: Visual Instruction Tuning with 500x Fewer Parameters through Modality Linear Representation-Steering" which explores highly parameter-efficient steering, or "Gradient-based Attribution for Inference-Time Steering of LLMs and VLMs," which provides a token-selective, attribution-guided alternative. The claim of architectural agnosticism and novelty is significantly weakened by this failure to rigorously compare and contrast against existing or very recent, conceptually similar cross-modal interventions.

2. Lack of Mechanistic Insight for Cross-Modal Transfer

The paper's key finding is a "novel cross-modal transfer" of textual steering vectors, yet the underlying mechanism is not explored beyond vague references to "preserved semantics" and "shared semantics." There is no deep analysis or experimental evidence to explain why a vector optimized to represent a concept in a purely linguistic context (e.g., in a text-only LLM backbone) so effectively manipulates the representation of a visual concept (e.g., spatial relationship in an image token) in the MLLM. The current evidence merely demonstrates correlation of effect, not causation or deep mechanistic understanding, leaving the central claim of the paper underspecified and unproven.

3. Instability and Quality Dependence of Steering Vector Extraction

The authors explicitly state that a "primary limitation" is the "reliance on the quality of extracted steering vectors" and that the vectors from SAE and Linear Probing "can be of poor quality and fail to adequately represent the target concepts," leading to "variable steering performance across different layers and models." This concession undermines two-thirds of the presented methodologies. The consistent superiority of MeanShift suggests the other two methods fail the core test of interpretability and effectiveness, rendering them more of a negative result rather than a robust cross-validation of the transfer effect. The current approach is therefore highly dependent on one specific, and less interpretable, extraction technique.

4. The Superior Generalization of Steering is Overstated and Task-Dependent

While the MeanShift method shows strong out-of-distribution (OOD) performance on highly focused, synthetic datasets like CLEVR (where it is significantly better than LoRA), its generalization advantage diminishes drastically on complex, real-world tasks. As the results in Table 5 show, the average improvement on real-world tasks like VQAv2, COCO Captions, and ChartQA is marginal (often $<+1.0\%$) and in many cases, non-significant or even negative. The claim of "superior generalization capabilities of steering" must be severely qualified: it is an effect primarily observed on datasets designed to isolate the exact visual concept being steered, not a general robustness property for complex, integrated multimodal reasoning.

5. Empirical Efficacy is Limited by Model Size and Prompting Baseline Weakness

The effectiveness of the steering intervention is inversely correlated with model size; the 3B model consistently shows a much larger gain than the 10B model, suggesting the approach is mainly effective for smaller, more malleable models. Furthermore, the prompting baseline is explicitly noted to "barely steer" the MLLMs, contrasting sharply with its efficacy in text-only domains. This weak baseline is a critical issue: the improvements observed may simply be a function of overcoming the inherent difficulty MLLMs have in following any fine-grained visual reasoning instructions, rather than proof of a uniquely powerful steering mechanism. The true margin over a stronger, more optimized baseline is unknown.

**Questions:**

See Above.

---

> ### Author Response · Authors · 2025-11-14
> **Author Response (Part 1)**
>
> We appreciate the reviewer’s detailed comments and address each point in turn. Several concerns appear to stem from overlooking parts of Sections 6-8 and Appendix C; we hope the clarifications below will resolve these.
>
> ### **W1: “Related work on cross-modal steering is underdeveloped; omits concurrent works.”**
>
> We acknowledge that multimodal steering is an emerging area with multiple concurrent approaches exploring different dimensions, and we agree to add more recent works beyond VTI. Although these works don’t necessarily contain interpretability insight, they definitely fall into the category of multimodal steering and deserve discussion in our related work.
>
> We would like to clarify that our contribution is complementary to these approaches— our novelty lies in **leveraging the shared representation space between the text-only backbone and the vision-finetuned model**. This specific insight enables reuse of existing interpretability infrastructure without requiring multimodal training data, and provides a plug-and-play mechanism where text-derived steering vectors transfer directly to multimodal contexts.
>
> Our contribution is not in being the first or only multimodal steering method, but rather in demonstrating this cross-modal transfer phenomenon through shared semantic representations.
>
> We will:
>
> * Expand the related-work section to include recent multimodal steering methods (including those exploring parameter-efficient interventions and gradient-based approaches), and
> * Clarify that our specific contribution is the text-to-vision transfer mechanism via shared representations in the backbone, enabling direct reuse of text-only interpretability tools.
>
> ### **W2: “No mechanistic insight; only vague ‘preserved semantics’ story.”**
>
> We agree that providing a mechanistic account of cross-modal transfer would be extremely valuable, and we are careful not to claim one. Our goal in this paper is primarily **empirical**: to demonstrate that this cross-modal transfer phenomenon exists and is robust.
>
> Regarding the concern about "causation": We respectfully note that our paper does not claim to establish causation in the sense of proving *why* the transfer occurs at a neural-circuit level. Rather, we provide substantial **behavioral evidence** that the transfer *does* occur and is systematic:
>
> - **Section 3** contains a color-perception steering experiment where an SAE feature associated with "red" in the text-only Gemma-2 backbone systematically modulates PaliGemma2's interpretation of a yellow-orange image. Crucially, as we vary steering strength, we observe smooth, semantically meaningful transitions (yellow → orange → red), not arbitrary or purely linguistic pattern-matching. This suggests the steering vectors operate on shared semantic representations rather than exploiting surface-level textual correlations.
>
> - **Sections 5-6** demonstrate that the same steering directions extracted from text-only backbones consistently improve performance on visual reasoning tasks (spatial relationships, counting) across multiple models and architectures. The steering vectors generalize to OOD benchmarks and real-world tasks, which would be unlikely if they merely captured linguistic artifacts rather than shared multimodal semantic features.
>
> We acknowledge that we do not fully explain the underlying mechanism of why shared representations enable this transfer. We will:
>
> * Clarify in the discussion that our contribution is demonstrating the robust existence and practical utility of this cross-modal transfer phenomenon, and
> * Explicitly highlight deeper mechanistic analysis (e.g., probing the geometry of shared representation spaces, identifying specific circuits) as important future work.

---

> > ### Author Response · Authors · 2025-11-14
> > **Author Response (Part 2)**
> >
> > ### **W3: “Instability of SAE / probe vectors undermines two-thirds of methods; they are negative results.”**
> >
> > We would like to clarify our intent: **we are not claiming to provide three perfect methods, but systematically comparing three established steering vector extraction techniques** to evaluate their cross-modal transfer effectiveness.
> >
> > The finding that MeanShift consistently outperforms SAE and Linear Probing is a **practical contribution, not a weakness**. For practitioners using our method to improve MLLM capabilities, they need at least one reliable extraction technique—and we demonstrate that MeanShift provides robust improvements across models, tasks, and OOD settings.
> >
> > Regarding "highly dependent on one specific, and less interpretable, extraction technique": MeanShift being the most efficient and straightforward method does not make it less interpretable than the other two. In fact, recent work (AxBench, Wu et al., 2025) shows MeanShift excels at concept extraction in text-only LLMs. Having one method that works best does not undermine the effectiveness of our approach—it identifies the most practical solution.
> >
> > We will revise the discussion to emphasize that identifying MeanShift as the most robust method is a positive finding for practitioners.
> >
> > ### **W4: “Superior generalization is overstated; gains on real-world tasks are marginal or negative.”**
> >
> > We appreciate this clarification and agree that our generalization claims should be more carefully scoped.
> >
> > Our claims about "strong generalization" are made only in the context of highly focused OOD benchmarks (Section 6.2 and Section 7) that isolate the exact visual reasoning skills we target—spatial relationships and counting. In these settings, steering maintains strong performance without hyperparameter tuning (+7.6 average improvement) while LoRA degrades significantly (+1.7 average).
> >
> > We explicitly acknowledge in Section 6.3 that improvement magnitudes are smaller on real-world tasks because "these complex tasks depend less exclusively on core spatial relation and counting skills that our steering vectors specifically target." We do not claim steering provides superior generalization for *all* multimodal reasoning—only for tasks heavily relying on the specific capabilities we enhance. The consistent improvements on real-world tasks (15/18 positive, 7 statistically significant) indicate that targeted capability enhancements can effectively transfer to complex applications.
> > To avoid overstating, we will further clarify the scope of our "superior generalization" claims to make explicit that they apply to capability-focused tasks rather than all multimodal reasoning scenarios.
> >
> > ### **W5: “Effectiveness limited by model size; prompting baseline is weak; margin over stronger baselines unknown.”**
> >
> > **Model size trend.**
> > We agree that steering is more impactful on the 3B model than on the 10B model, and we view this as an interesting empirical finding rather than a weakness:
> >
> > - On CV-Bench (Table 1), PaliGemma2-3B often sees larger absolute gains than PaliGemma2-10B.
> > - However, the 10B model still benefits—for example, MeanShift yields **+4.0** relation accuracy on CV-Bench and up to **+9.8** improvements on CLEVR counting.
> >
> > We will clarify that the method remains effective for larger models, although smaller models appear more “steerable,” which is consistent with the idea that larger models may already encode stronger priors and be somewhat less malleable.
> >
> > **Prompting baseline.**
> > Our prompting baseline is not intentionally weak. We follow the same rigorous grid search protocol used in text-only steering work (AxBench), generating 96 candidate prompts per taxonomy and selecting the best-performing one on held-out data. The prompting baseline represents the ceiling of what can be achieved by exploiting MLLMs' textual instruction-following alone. The fact that prompting performs poorly while our steering methods show substantial improvements indicates that our approach enhances true visual reasoning capabilities rather than merely exploiting linguistic patterns. Moreover, to our knowledge, there are no other interpretable, training-free steering methods suitable as baselines for multimodal models.
> >
> > Finally, we **do** compare against a stronger supervised baseline, LoRA finetuning, in Section 7, where LoRA achieves higher in-distribution performance than steering, but shows much weaker OOD gains than our steering method, despite training overhead and loss of interpretability.

---

> > > ### Comment · Reviewer_qPh5 · 2025-11-25
> > >
> > > Thanks for your reply. Given for the steering method, I am more interested in understanding the underlying mechanisms than in empirical results. I will revise my rating.

---

### Official Review · Reviewer_4Qp8 · 2025-10-29

**Soundness:** 2
**Presentation:** 2
**Contribution:** 2
**Rating:** 4
**Confidence:** 4

**Summary:**

This paper proposes a novel approach to steer Multimodal Large Language Models (MLLMs) by using "steering vectors" derived from their text-only LLM backbones. The central finding is that semantic representations from the text-only LLM remain effective for intervention even after multimodal fine-tuning, allowing for lightweight, inference-time activation additions to enhance specific visual capabilities like spatial reasoning and counting. The authors demonstrate that this method can achieve better out-of-distribution generalization compared to LoRA fine-tuning. The paper suffers from significant limitations in its experimental setup, evaluation scope, and practical utility, which collectively undermine the generality and impact of its conclusions.

**Strengths:**

1. The key contribution is demonstrating that representation steering from the text domain can be transferred to the multimodal domain to improve visual task performance.
2. The proposed method is a "plug-and-play" technique that does not require any model retraining or fine-tuning.

**Weaknesses:**

1. The evaluation is confined to PaliGemma2 and Idefics3-8B. It fails to include a representative set of more recent, powerful, and widely-used open-source MLLMs, such as the LLaVA series, the Qwen-VL family, or InternVL series. This narrow selection makes it difficult to ascertain whether the findings are a generalizable phenomenon or an artifact of the specific architectures tested. Without broader validation, the claims of general applicability are unsubstantiated.
2. The paper's evaluation is heavily concentrated on a few sub-tasks from CV-Bench (spatial relations, counting) and closely related OOD datasets. These are diagnostic, often simplified tasks designed to probe isolated skills. The method's utility is largely confined to simple, isolated cognitive skills and does not translate effectively to the complex, compositional reasoning required by general-purpose multimodal applications.
3. The method requires a user to first define a specific capability to improve (e.g., counting, color recognition), then extract a corresponding steering vector, and finally perform an expensive grid search to find the optimal intervention layer and strength for each model-task pair. This workflow is cumbersome and does not scale to open-ended, complex scenarios where the required "cognitive skills" cannot be easily enumerated and calibrated in advance.

**Questions:**

Sea weakness.

---

> ### Author Response · Authors · 2025-11-14
>
> We thank the reviewer for recognizing the transfer of text-domain steering to multimodal models and the plug-and-play nature of our approach.
>
> ### **W1: "Model coverage limited to PaliGemma2 and Idefics3; unclear generality."**
>
> Our selection of PaliGemma2 (3B, 10B) and Idefics3 (8B) was driven by our research goal: **systematically comparing three steering vector extraction methods** (SAEs, Mean Shift, and Linear Probing). This requires high-quality, comprehensive SAEs—we use **GemmaScope** for Gemma-2 and **LlamaScope** for Llama-3.1, currently the only publicly available SAEs with sufficient coverage for our comparison.
>
> Qwen-VL, InternVL, and other LLaVA variants use backbones (Qwen, InternLM, Mistral) lacking comparable SAEs. We could apply Mean Shift and Linear Probing to these models, but would lose SAE-based steering comparison. Despite this constraint, our models span diverse architectures (prefix-LM vs. fully autoregressive) and scales (3B, 8B, 10B) with consistent effectiveness. We agree that including additional families such as Qwen-VL or InternVL would further support generality; we view this as future work contingent on having SAEs for their backbones.
>
> We will revise the text to better explain our model selection as enabling systematic comparison of extraction methods, and acknowledge the constraint of our evaluation.
>
> ### **W2: "Utility confined to simplified CV-Bench tasks; not useful for real multimodal applications."**
>
> Our evaluation on real multimodal applications is included in Section 6.3/Appendix C (Table 5), which may have been overlooked. We evaluate on six diverse real-world tasks: VQAv2, COCO Captions, DocVQA Layout/Number, ChartQA, and VTabFact. Using the **same** CV-Bench-tuned steering vectors, MeanShift improves **15 out of 18** model-task combinations with several significant gains (up to +6.0 DocVQA Layout, +4.6 COCO Captions, +4.4 ChartQA, +5.5 VTabFact). All methods show consistent positive average gains.
>
> This transfer from simple CV-Bench-tuned vectors to complex real-world applications demonstrates that textual steering captures generalizable visual reasoning capabilities. We acknowledge that improvements on these complex tasks are smaller than on capability-focused benchmarks (e.g., CLEVR's +19.6 average for counting)—this is expected and explicitly noted in Section 6.3, as these tasks depend on many skills beyond the single steered capability.
>
> We will: (1) Move Table 5 and the real-world task analysis from Appendix C to the main text for better visibility. (2) More prominently describe these evaluations as complex multimodal tasks so the broader evaluation scope is immediately clear to readers.
>
> ### **W3: “Workflow (define skill → extract vector → grid search) is cumbersome and not scalable.”**
>
> We agree that a naive, per-task grid search would be cumbersome; our actual procedure is more lightweight and amortized:
>
> **Grid search is small and performed once per capability:**
> - Layers are restricted to middle layers where visual information is concentrated (e.g., {5,…,20} for PaliGemma2-3B)
> - Scale factors are drawn from small sets (6 values per method)
> - For a given capability (e.g., "spatial relations") and model, we tune ($\ell$, $\alpha$) **only once** on the CV-Bench training split
> - The selected ($\ell$*, $\alpha$*) are then **reused across all** OOD datasets and real-world tasks that depend on that capability with no per-dataset re-tuning
>
> On the other hand, finetuning like LoRA requires hyperparameter selection (rank, learning rate, etc.) and full training runs per task, which is much more computationally intensive than our inference-time interventions. Moreover, as shown in Table 3, LoRA achieves poor OOD generalization (+1.7 average), meaning **separate training would be required for each new task or distribution**. In contrast, our steering vectors generalize effectively without retraining.
> **Regarding open-ended scenarios:** Our work deliberately targets atomic visual capabilities (spatial relations, counting, attributes, entities) that are well-established in the vision-language literature as fundamental building blocks of visual understanding. For applications requiring these capabilities, our approach provides an efficient alternative to task-specific fine-tuning. Importantly, improving these atomic capabilities also improves model performance on complex scenarios that are not solely dependent on these capabilities, as shown in Section 6.3. We acknowledge that extending to richer capability libraries or developing meta-methods for automatically selecting ($\ell$, $\alpha$) would be valuable future work, and we will state this explicitly in the revision.

---

### Official Review · Reviewer_kK1x · 2025-11-01

**Soundness:** 3
**Presentation:** 2
**Contribution:** 3
**Rating:** 4
**Confidence:** 3

**Summary:**

This paper presents an interesting insight: steering vectors derived from LLM backbones can effectively guide their multimodal counterparts to enhance visual understanding. The authors systematically validate the effectiveness of various steering vector extraction methods across different MLLM architectures and visual reasoning tasks. Experimental results demonstrate that textual steering vectors significantly improve MLLMs' performance on tasks such as spatial relation reasoning and object counting.

**Strengths:**

1. The paper presents a novel insight by demonstrating the transferability of textual representations from LLMs to MLLMs, revealing an effective steering mechanism.
2. The method achieves strong empirical performance, consistently improving results on spatial relation and counting tasks across multiple MLLM architectures.
3. The paper is well-written and easy to follow.

**Weaknesses:**

1. The evaluation is limited to visual reasoning tasks (e.g., counting, spatial relations) without exploring more complex multimodal scenarios such as OCR.
2. The experiments are restricted to a small set of MLLM backbones; validation on more recent and advanced models (e.g., Qwen-VL, InternVL families) would further strengthen the generality of the conclusions.

**Questions:**

See Weakness

---

> ### Author Response · Authors · 2025-11-14
>
> We thank the reviewer for the positive assessment of the novelty, empirical strengths, and writing.
>
> ### **W1: "Evaluation is limited to visual reasoning tasks… no OCR-like complex scenarios."**
>
> Our evaluation already includes complex tasks beyond simple visual reasoning, but they appear in Section 6.3/Appendix C (Table 5), which may have been overlooked. We evaluate on six real-world multimodal tasks:
> * VQAv2 (general VQA),
> * COCO Captions (open-ended captioning),
> * DocVQA Layout and DocVQA Number (document understanding and OCR),
> * ChartQA (chart reasoning), and
> * VTabFact (table reasoning).
>
> Crucially, using the same spatial-relation or counting steering vectors (tuned only on CV-Bench), MeanShift improves performance in 15 out of 18 model-task pairs on these real-world tasks, with several statistically significant gains. For instance:
> * DocVQA Layout: up to +6.0 points,
> * COCO Captions: up to +4.6 points,
> * ChartQA: up to +4.4 points, and
> * VTabFact: up to +5.5 points in individual settings, with all positive average gains.
>
> This transfer from simple CV-Bench-tuned steering vectors to complex real-world tasks strengthens our claim that textual steering vectors capture generalizable visual reasoning capabilities.
>
> We will revise the manuscript to:
> 1. Move Table 5 and the real-world task analysis from Appendix C to the main text to ensure these results receive appropriate visibility.
> 2. More prominently describe these evaluations as complex multimodal tasks throughout the paper, rather than only "real-world benchmarks," so the broader scope is immediately clear to readers.
>
> ### **W2: "Experiments are restricted to few MLLMs; include Qwen-VL, InternVL, etc."**
>
> We appreciate the suggestion to expand model coverage. Our selection of PaliGemma2 (3B, 10B) and Idefics3 (8B) enables **systematically comparing three steering vector extraction methods** (SAEs, Mean Shift, Linear Probing), which is central to our contribution. These models have high-quality, comprehensive SAEs for their backbones (GemmaScope for Gemma-2, LlamaScope for Llama-3.1), allowing fair comparison across all three methods.
> **Regarding Qwen-VL, InternVL, and other LLaVA variants:**
>
> These models use language backbones (Qwen, InternLM, Mistral, etc.) that lack comprehensive, publicly available SAEs comparable to GemmaScope/LlamaScope. While some partial SAEs exist (e.g., for Qwen), they do not cover all layers needed for our systematic analysis. We could apply Mean Shift and Linear Probing to these models, but would lose the ability to compare against SAE-based steering, making the method comparison incomplete.
>
> **Generalization across architectures and broader applicability:**
>
> Our results demonstrate effectiveness across different model sizes (3B, 8B, 10B) and two architecturally distinct MLLMs (prefix-LM PaliGemma2 vs. autoregressive Idefics3), suggesting the approach generalizes across architectural choices. The systematic comparison across extraction methods reveals that Mean Shift achieves the most consistent performance, providing practical guidance for future applications.
>
> **We will clarify in the revision:**
> 1. Our model selection enables systematic comparison of extraction methods, which is a core contribution.
> 2. Our steering framework (Mean Shift and Linear Probing, and SAE when available) can be applied to any MLLM with an LLM backbone (including Qwen-VL, InternVL, LLaVA variants), and extending our evaluation to these models is valuable future work.

---

### Note · Authors · 2025-12-29

**Comment:**

We have decided to withdraw our submission to improve the work and address the reviewers' concerns.

**Withdrawal Confirmation:**

I have read and agree with the venue's withdrawal policy on behalf of myself and my co-authors.